

# Estimating regional scale methane flux and budgets using CARVE aircraft measurements over Alaska

Sean Hartery[1], Róisín Commane[2], Jakob Lindaas[2], Colm Sweeney[3,4], John Henderson[5], Marikate Mountain[5], Nicholas Steiner[6], Kyle McDonald[6], Steven J. Dinardo[7], Charles E. Miller[7], Steven C. Wofsy[2], and Rachel Y.-W. Chang[1,2]

[1]Department of Physics and Atmospheric Science, Dalhousie University, Halifax NS
[2]School of Engineering and Applied Sciences, Harvard University, Cambridge MA
[3]Global Monitoring Division, National Oceanic and Atmospheric Administration Earth System Research Laboratory, Boulder CO
[4]Cooperative Institute for Research in Environmental Sciences, University of Colorado, Boulder CO
[5]Atmospheric and Environmental Research, Inc., Lexington, MA
[6]Department of Earth and Atmospheric Science, City College University of New York, New York NY
[7]Jet Propulsion Laboratory, California Institute of Technology, Pasadena CA

*Correspondence to:* R.Y.-W. Chang (rachel.chang@dal.ca)

**Abstract.** Methane ($CH_4$) is the second most important greenhouse gas but their emissions from northern regions is still poorly constrained. In this study, we analyze a subset of in situ $CH_4$ aircraft observations made over Alaska during the growing seasons of 2012–2014 as part of the Carbon in Arctic Reservoir Vulnerability Experiment (CARVE). Surface $CH_4$ fluxes are estimated using an atmospheric particle transport model which quantitatively links surface emissions from Alaska and the western Yukon

with observations of enhanced $CH_4$ in the boundary layer. We estimate that between May and September, 2.1±0.5 Tg, 1.7±0.4 Tg and 2.0 ±0.3 Tg of $CH_4$ were emitted from the region of interest for 2012–2014, respectively. The predominant sources of the $CH_4$ budget were two broadly classed eco-regions within our domain, with $CH_4$ from the tundra region accounting for over half of the overall budget, despite only representing 18% of the total surface area. Boreal regions, which cover a large part of the study region, accounted for the remainder of the emissions. Simple multiple linear regression analysis revealed that

overall, $CH_4$ flux were largely driven by soil temperature and elevation. In regions specifically dominated by wetlands, soil temperature and moisture at 10 cm depth were important explanatory variables while in regions that were not wetlands, soil temperature and moisture at 40 cm depth were more important, reflecting the depth at which methanogenesis occurs. Although similar variables have been found in the past to control $CH_4$ emissions at local scales, this study shows that they can be used to generate a statistical model to estimate the regional scale $CH_4$ budget.

# 1  Introduction

Recent trends in observed global atmospheric methane ($CH_4$) mole fractions have shown increases since a short-lived stabilization period in the early 2000's and have increased by ∼150% from pre-industrial values (Dlugokencky et al., 2011; Kirschke et al., 2013; Ciais et al., 2014). As the second most potent anthropogenically-emitted greenhouse gas after carbon dioxide



$(CO_2)$ in terms of total radiative forcing, $CH_4$ can account for 20% of recent trends in global surface air temperatures, which have risen approximately 0.6 K over the past century (Kirschke et al., 2013; Ciais et al., 2014). Since emissions of $CH_4$ from wetlands represent ~30% of the global $CH_4$ produced annually, it is of critical scientific interest to determine whether these sources will strengthen in a warming climate (Whalen, 2005; Kirschke et al., 2013; Ciais et al., 2014). In particular, it has

been speculated that increased air temperatures in wetlands north of 40N, combined with ecological, hydrological, and bio-geochemical changes could couple into a temperature-emissions feedback, potentially leading to a depletion of organic carbon previously sequestered in below-ground permafrost and increased atmospheric $CH_4$ (Schuur et al., 2015; Tarnocai et al., 2009).

Temperature increases have been more drastic in Arctic regions, which have seen almost 3 K increases in air temperatures since the beginning of the 20[th] century (Overland et al., 2015), with 1.8 K occurring over the past three decades (Collins

et al., 2013). Rising air surface temperatures have resulted in a response in soil temperatures, with winter-time observations of permafrost core temperatures in Alaska showing increases of approximately 3–4 K on the Arctic Coastal Plain of Alaska at borehole depths of 5–20 m and 1–2 K in the Brooks Range at depths of 20 m (Osterkamp, 2005). If current Arctic climate trends continue, up to 10–30% of permafrost in Arctic lowlands could significantly degrade, leading to measurable ecological shifts and adding new labile organic carbon to the carbon cycle (Jorgenson et al., 2006).

From a biogeochemical perspective, $CH_4$ is produced in soils below the water table, which provide the anaerobic conditions necessary for fermentation of soil organic carbon stocks (Whalen, 2005). The fermented organic carbon products are then consumed by methanogenic archaea within the soil column, producing $CH_4$ gas (Whalen, 2005). As $CH_4$ production is a biological process relying on microbial activity, it is commonly observed that high $CH_4$ emissions are coincident with warm, wetland soils (Sturtevant et al., 2012; Olefeldt et al., 2013; Kirschke et al., 2013; Christensen, 1993).

In Alaska and the neighbouring Yukon, seasonal wetlands make up nearly 12% of the total surface area (Bergamaschi et al., 2007), with 92% of soils within continuous permafrost zone (Hugelius et al., 2013). Since Alaska is frozen most of the year, its carbon stocks have long been preserved within permafrost, limiting carbon mobilization through respiration. In spite of recent warming and mobilization of sequestered carbon, a recent study of atmospheric mole fractions of $CH_4$ in the North Slope of Alaska found no significant increase in $CH_4$ emissions over the past 29 years (Sweeney et al., 2016). Field observations have

also reported that microbial communities linked to $CH_4$ oxidation thrive in soils with low moisture content (Xue et al., 2016), highlighting that a warming climate may not have a one-to-one effect on biogenic $CH_4$ flux at a regional scale.

Numerous past studies have been conducted in permafrost regions such as Alaska at the scale of chambers (as summarized by Olefeldt et al., 2013) and eddy-covariance towers (e.g. Fan et al., 1992; Sturtevant et al., 2012; Zona et al., 2016), providing insight on factors controlling $CH_4$ emissions at the scale of ~1 m to ~1 km. These studies have revealed that $CH_4$ emissions

are spatially inhomogeneous at those scales and are highly dependent on local conditions such as soil moisture, temperature, elevation and soil carbon. While these process-based studies are extremely important, extrapolating the results to larger scales can be challenging. At the other extreme, top-down inversion studies estimate global $CH_4$ emissions using measurements from surface sites around the world and/or satellite observations coupled to sophisticated transport models (Bruhwiler et al., 2014; Bergamaschi et al., 2013; Chen and Prinn, 2006). These results provide insight on regional emissions; however, it is more

difficult to understand local drivers that affect emission rates. More recently, tall towers, either alone or in a network, and



aircraft observations, have been used to study regional emissions of $CH_4$ from permafrost areas (Karion et al., 2016; Sasakawa et al., 2010; Chang et al., 2014). Tall towers are advantageous since their operation is less dependent on weather conditions than static chamber measurements, they provide continuous measurements and their footprint spans a much larger area than a typical eddy-covariance tower. Despite this, in a region like Alaska, where high mountains significantly affect transport

patterns, a single tower may not be sensitive to the entire region throughout the year (Karion et al., 2016). In contrast, aircraft observations, by virtue of their mobile platform, can sample larger regions and can periodically measure in the free troposphere to establish background levels. However, their coverage is dependent on weather conditions.

In this study, we estimate surface $CH_4$ fluxes using in situ observations of $CH_4$ from an aircraft that flew in Alaska as part of the Carbon in Arctic Reservoirs Vulnerability Experiment (CARVE). This work uses similar methods as Chang et al. (2014),

but extends the analysis to include the growing seasons of 2013 and 2014 and explores how their interannual and intra-annual variability can be explained by hydrological and environmental controls at a regional scale. A recent study by Miller et al. (2016) explores similar questions using a more complex geostatistical inversion model constrained by a much larger data set. Using simple multiple linear regression models, we investigate the relationship between land surface properties and observed atmospheric $CH_4$ and find that the results are similar. This could provide a simpler diagnostic tool for regional carbon cycle

analysis.

## 2 Methods

### 2.1 Description of aircraft flights

The data presented in this study were collected on board a National Aeronautics and Space Agency (NASA) C23-B aircraft during the Alaskan growing seasons of 2012–2014. In 2012, flights occurred in the last two weeks of each month from May

to September for a total of 31 flight days and 212 flight hours. In 2013 and 2014, flights occurred in the first two weeks of each month from April to October and May to November, resulting in 290 and 300 flight hours over 42 and 48 flight days, respectively. All flights analyzed in this study originated from Fairbanks, AK and stayed within Alaska. Not all regions and altitudes were sampled every month since the flight routes were limited by icing concerns and weather conditions. Data were typically acquired at 150 m above ground level (AGL) to maximize sensitivity to local surface-atmosphere fluxes; however,

periodic profiling of the atmosphere to 5–6 km AGL occurred throughout the flights to characterize planetary boundary layer height and mole fractions in the free troposphere.

### 2.2 Measurements

Mole fractions of $CH_4$, $CO_2$, carbon monoxide (CO) and water vapour ($H_2O$) were measured in situ every ~2.5 s using two independent cavity ring-down spectrometers. Each system had an individual rear-facing inlet on the port side of the

aircraft. Sample flow from the inlets passed through a length of Synflex tubing with an approximate transit time of $30\pm2$s. The first system (Picarro; G1301-m in 2012, G2401-m in 2013 and 2014) directly sampled the air without any pre-treatment and





alternately sampled from one of two on-board calibration cylinders every 30 min. These on-board calibration cylinders were changed throughout the campaigns as the pressure approached 3.4 MPa (500 psi). They were calibrated by NOAA before and after each deployment. Calibration for the water vapour correction was conducted on the flight instrument before and after each year's campaign according to Chen et al. (2010). Further details of this system can be found in Karion et al. (2013).

The sample flow in the second system first passed through a 0.2 $\mu$m Teflon filter before passing through a Nafion dryer followed by a dry ice trap which effectively lowered the dewpoint temperature of the sample flow to ~195K before entering the spectrometer (Picarro; G2401-m). This reduced the water vapour of the sample flow to $< 0.001\%$ and allowed dry mole fractions of $CH_4$, $CO_2$ and CO to be directly measured. Two onboard 8 L calibration cylinders (independent of the first system's) were sampled at the beginning and end of every flight as well as every 30 min during flight. These in-flight calibrations

were linearly interpolated between calibration times, which were then additionally interpolated to calculate the measured mole fraction in between calibrations. In-flight calibrations that were greater than 2.5 standard deviations away from the mean of the calibrations for a given flight were excluded.

Before each year's campaign, the onboard calibration cylinders for the second system were flushed twice and then filled from 30 L fill tanks (Scott Marrin). Due to the longer sampling seasons in 2013 and 2014, the onboard calibration tanks were topped

up once in each of those years with the original fill tanks before the pressure dropped below 3.4 MPa. Both the original fill tanks from Scott Marrin and the onboard calibration cylinders were calibrated in the laboratory with the spectrometer used in flight before and after each year's campaign using tanks with known mole fractions obtained from NOAA, tying our measurements to the WMO scales (Dlugokencky et al., 2005; Novelli et al., 1994; Zhao and Tans, 2006). The differences in mole fractions before and after each year's flight campaigns in both the on-board calibration cylinder and the fill tanks were less than 0.15

ppm for $CO_2$, 0.8 ppb for $CH_4$ and 7 ppb for CO, all of which were within the precision of the system. The mole fraction in the on-board calibration cylinders was therefore treated as constant throughout each year's study and assumed to be unaffected by the addition of gas. It was also assumed that any changes within any of the tanks were negligible throughout each year's campaign.

The one deviation was that the onboard calibration cylinders were not calibrated before the 2012 deployment due to time

constraints. However, these cylinders were calibrated after the 2012 campaign and compared to the fill tanks calibrated before and after that year's campaign. Except for $CO_2$ in one of the onboard calibration cylinders, which was 0.29 ppm lower than the fill tank, the comparison with the fill tanks for the other gases all fell into the ranges given above. In the case of this exception, the post-mission calibration of the onboard calibration cylinder was used since the tank was not topped up during that year's mission.

Comparison of the two systems showed a mean difference of 0.8, 0.3 and 0.4 ppb for $CH_4$ for 2012–2014, respectively with no dependence on water vapour levels ($r^2 = 0.008$). The mole fractions presented in this study merges those measured by both systems, to fill in times when one of the systems was calibrating or functioning imperfectly. The system used to gap-fill is offset by the mean difference between the two systems for a given flight. Since our analysis is based on the difference between mole fractions in the free troposphere and the mixed layer (see below), this treatment does not bias our analysis. The greenhouse

gas measurements were merged to a common 5 s time scale along with the location data, measured using a global positioning





unit (GPS) (Crossbow; NAV420); outside air pressure (Parascientific; 745-15A); outside air temperature (Harco; 100366-18), dewpoint temperature (Edgetech; Vigilant) and ozone (2B Technologies; 205).

## 3 Analysis methods

### 3.1 Footprint sensitivity

To relate the variation of observed mole fractions of $CH_4$ to surface processes, an adjoint atmospheric transport model was used to simulate the sensitivity of the air parcels measured by the aircraft to the surface influence. Specifically, the Weather Research and Forecasting (WRF; Skamarock et al. (2008)) regional numerical weather prediction model was coupled offline to the Stochastic Time-Inverted Lagrangian Transport model (STILT; Lin et al. (2003)). The coupling of the STILT model with WRF meteorological fields, hereafter WRF-STILT, is described by Nehrkorn et al. (2010) The polar variant of WRF (Hines et al., 2011; Bromwich et al., 2009; Hines and Bromwich, 2008) coupled with STILT was configured for the Arctic CARVE domain by Henderson et al. (2015), who provide a detailed description and meteorological validation of the high-resolution simulations performed on a 3.3-km grid suitable for simulating particle transport on regional scales in mountainous terrain. Their initial STILT footprints from WRF v3.4.1 were used in the analysis of Chang et al. (2014). The current investigation uses WRF v3.5.1 and incorporates recently-available PIOMAS cryosphere fields (Zhang and Rothrock, 2003; Hines et al., 2015) and footprints on an expanded circumpolar domain north of 30N, among other methodological improvements.

To derive source-receptor relationships, each STILT simulation launched 500 particles into the atmosphere at the location of a measurement and traced its dispersion through the atmosphere in reverse time over 10 days via stochastic (i.e. random/probabilistic) processes determined from the underlying meteorology (Lin et al., 2003; Henderson et al., 2015). At every one-hour time step, particles that are in the lower half of the boundary layer are assumed to be influenced by the surface and are gridded to generate a footprint sensitivity plot. The footprint used in this analysis was calculated on a $0.5° \times 0.5°$ lat/lon grid. The unit of the footprints is abundance $flux^{-1}$ (e.g. ppb $(nmol\ m^{-2}s^{-1})^{-1}$, where ppb is parts per billion) (Lin et al., 2003) and allows us to relate surface emissions to an observation at a given time and location.

To avoid unnecessary computational costs, the measurement locations were first placed into discrete bins on a grid of 5 km in the horizontal, 50 m in the vertical below 1 km and 100 m in the vertical above 1 km (Chang et al., 2014). Measurement times were also rounded to the hour to match the time resolution of the meteorological fields. The use of WRF at high resolution to drive the STILT model results in higher fidelity meteorological fields and subsequent transport calculations than afforded by most global reanalysis (Nehrkorn et al., 2010; Henderson et al., 2015).

### 3.2 Domain

This analysis focuses on the influence of surface emissions from Alaska and Yukon on observed $CH_4$ mole fractions. To this end, we restricted our region of interest to 50–75N and 130–170W (Fig. 1) and only consider footprint sensitivity derived from WRF-STILT during the first five days preceding each receptor point. The time-scale of five days was chosen under the





assumption that sources outside the domain would have transport timescales roughly equal to that of free-tropospheric mixing and would therefore only contribute to background values of $CH_4$. Figure 2 shows the cumulative 5 day footprint of our analyzed times for each year averaged over the number of profiles in each year, and shows that our observations are most influenced by the boreal interior of Alaska as well as tundra regions. It should be noted that although our mean footprints were

more sensitive to the North Slope of Alaska in 2012, our absolute sensitivity was comparable in all three years due to the increased number of flights in 2013 and 2014.

Our choice of spatial domain was determined by the distribution in five-day footprint. As will be defined in the subsequent section concerning the mixed layer, footprint sensitivity simulations that had less than 1.0 ppb (nmol m$^{-2}$ s$^{-1}$)$^{-1}$ total land sensitivity were identified as being in the free troposphere, following Henderson et al. (2015). For an idealized homogeneous

distribution of sensitivity, where an air parcel is equally sensitive to all land areas within the domain, this threshold would translate to $\sim$1$\times$10$^{-4}$ ppb (nmol m$^{-2}$ s$^{-1}$)$^{-1}$ in any given grid cell. It was assumed that areas of the globe with sensitivities smaller than this threshold did not significantly contribute to $CH_4$ enhancements, since, even at extreme fluxes of 1000 mg m$^{-2}$ d$^{-1}$, they would contribute <0.1 ppb to the observed enhancement. In the cumulative 5 day footprints shown in Fig. 2, 71.2% of the footprint that exceeded the grid-cell threshold was from land in the domain of choice, while 28.5% could be

sourced to oceans. It is worth noting that the previous study of Chang et al. (2014) restricted their domain to 135°W. While the May–September footprints showed that <2% of the footprint originated in regions east of 135°W, it was found that at least 5% of the total land sensitivity in April, October, and November came from the Canadian Yukon. Of all the land surface influence on our observations, only 0.3% originated from land outside of our study region, suggesting that our sampling strategy and choice of domain allowed our observations to be most sensitive to surface emissions from the study region.

## 3.3   Mixed layer

We use the term 'mixed layer' to refer to the combination of both the residual layer and the boundary layer throughout the remainder of this paper. As our footprint sensitivity predicts that we are sensitive to surface fluxes 5 days previous to measurement, quantifying the total amount of $CH_4$ emitted requires calculating the amount of $CH_4$ in both the boundary layer and the residual layers from previous days. By determining the altitude at which the residual layer is capped, it is possible to determine

the portion of a vertical profile that is affected by surface emissions versus the portion that represents background levels in the free troposphere. The $CH_4$ column enhancement within the mixed layer will be used in Sect. 3.4 to estimate $CH_4$ emissions.

We estimate the bottom of the free troposphere ($h$) by calculating the refractivity ($N$) using parameters measured at different altitudes on the aircraft (Chan and Wood, 2013; Bean and Dutton, 1966):

$$N = 77.6\frac{P}{T} + 3.77 \cdot 10^5 \frac{P_w}{T^2}, \tag{1}$$

where $P$ and $P_w$ are the atmospheric pressure of air and water in hPa, respectively, and $T$ is the atmospheric temperature in K. The height at the minimum of the gradient in $N$ is the top of the mixed layer.

In analyzing individual profiles it was often found that this method over-estimated the boundary layer height as compared to profiles of $CH_4$, $O_3$ and the total land surface sensitivity calculated by WRF-STILT; as a secondary calculation, the height





at which WRF-STILT sensitivities dropped below 1.0 ppb (nmol m$^{-2}$ s$^{-1}$)$^{-1}$ was used to approximate $h$ (Henderson et al., 2015). Upon individual inspection of profiles it was found that in 25% of cases (i.e. 68 profiles) neither method captured an $h$ which correctly separated the free troposphere and mixed layer. After removing these profiles, a further 25% of profiles within the remaining set (i.e. 50) had $h$ differing by >750 m between the two methods (paired t-test; p<0.005). In these cases, either

the boundary layer dynamics within WRF-STILT were not representative of local meteorology or there was too much variance about the minimum of the gradient in $N$. Therefore, these profiles were not included in the analysis. In the final set of profiles used for analysis, a paired t-test found that the two methods did not statistically differ from each other (p>0.1) and both agreed to within 500 m $\sim$90% ($r^2$ = 0.6) of the time. We attempted to use other variables such as virtual potential temperature to define the mixed layer but the methods described above gave the most consistent results.

Across all years $h$ was found to occur between 1.1 and 1.8 km with a median of 1.5 km above Alaska, consistent with estimates derived from satellite retrievals (Chan and Wood, 2013). On average, $h$ in May–September was 30% higher than in April, October or November. These findings are consistent with lower solar zenith angles and shorter solar days in colder months, both of which reduce the strength of convective forces which form the boundary layer (Stull, 1988).

### 3.4 CH$_4$ flux estimates

In this analysis, we use column enhancements of CH$_4$ ($\Delta$CH$_4$) to estimate surface fluxes (Chang et al., 2014; Gatti et al., 2014; Chou et al., 2002). This method assumes that CH$_4$ enhancements below $h$ that are above background levels are a result of interactions with the local surface that are not well-mixed with the free troposphere. Surface fluxes are determined by calculating the corresponding surface influence throughout the column using the footprint sensitivities determined from WRF-STILT. This method of calculating $\Delta$CH$_4$ is less reliant on the accurate simulation of the vertical structure of the atmosphere as

well as turbulent transport in the lower atmosphere. Instead, we rely on the integrated model simulation to match the integrated observations.

To calculate $\Delta$CH$_4$, measurements in a vertical profile were first block-averaged by altitude into 250 m bins. A sample of such a profile observed during the CARVE campaign is shown in Fig. 3 and shows median, minimum and maximum [CH$_4$] observed in each 250 m bin up to 3 km along with the estimated $\Delta$CH$_4$ from the WRF-STILT footprint influence according to

Eq. 3. In this case the calculation of the refractive index ($N$) properly captured the transition from the mixed layer to the free troposphere and WRF-STILT adequately modelled the mixed layer, leading to a reasonable estimate of CH$_4$ flux. Using $h$ as the upper bound on the section of the profile affected by local sources and sinks:

$$\Delta\text{CH}_4 = \int\limits_0^h ([\text{CH}_4](z) - [\text{CH}_4]_0) \frac{P_d(z)}{R^* T(z)} dz, \qquad (2)$$

where $\Delta$CH$_4$ is the column enhancement in the mixed layer of CH$_4$ in (ppb), [CH$_4$]$_0$ is the estimated background of CH$_4$

in the free troposphere averaged in the first kilometer above $h$, $P_d$ is the calculated atmospheric pressure of dry air using measurements of $P$ and $P_w$, $R^*$ is the universal gas constant, and $z$ is the height above ground level (AGL) as calculated from the height above sea level measured by the GPS and an underlying digital elevation map within the input meteorology of WRF.





As an entrainment zone was usually evident in profiles of $CH_4$, where mixing ratios were between free troposphere and mixed layer values, 500 m was added to mixing layer height in order to estimate $[CH_4]_0$. Since $[CH_4]_0$ is stable with altitude in the free troposphere, this should not affect the integral in Eq. 2.

A similar calculation was undertaken for the corresponding footprint sensitivity to calculate the sensitivity of the mixed layer to the total surface ($\Delta I$ with units of ppb (nmol m$^{-2}$ s$^{-1}$)$^{-1}$). As a first estimate, our a priori $CH_4$ flux is assumed to be uniformly emitting over our entire domain. Footprint sensitivities were restricted to the study domain and masked from seas and mountains, which we assume to be neither sources nor sinks of atmospheric $CH_4$ and therefore did not contribute to $\Delta CH_4$. The total mixed layer sensitivity, $\Delta I$, is then scaled to match $\Delta CH_4$. This scaling factor is the $CH_4$ flux estimate, $\hat{E}$:

$$\hat{E} = \frac{\Delta CH_4}{\Delta I}, \tag{3}$$

with appropriate unit conversion to mg $CH_4$ m$^{-2}$ d$^{-1}$.

Monthly mean $CH_4$ flux ($\overline{E}$) were estimated by weighing individual flux estimates by the total footprint sensitivity of each profile (Chang et al., 2014; Karion et al., 2016), so that:

$$\overline{E} = \frac{(\sum \hat{E} \times \Delta I)}{\sum \Delta I} = \frac{\sum \Delta CH_4}{\sum \Delta I}. \tag{4}$$

This allowed profiles with larger spatial coverage to be more represented while profiles with lower $\Delta I$, which sometimes resulted in a high $\hat{E}$, did not inadvertently skew the mean estimate.

Growing season budgets were calculated by integrating the monthly emissions from May through September and over our study domain (excluding mountains). Uncertainties in this budget are estimated using the standard error calculated from the weighted standard deviations in the monthly means. The surface area of our domain was estimated using a digital elevation map from the Advanced Spaceborne Thermal Emission and Reflection Radiometer (ASTER) to estimate the enhancement of surface area in sloped terrain (Jenness, 2004). A land/ocean mask at $0.1°$x$0.1°$ was then used to approximate the fraction of land cover at $0.5°$x$0.5°$ to estimate surface area in coastal grid cells (i.e. $A_{land} = A^* f_{land}$). The total land surface area of our domain is estimated to be 2.1 million km$^2$ including mountains and 1.04 million km$^2$ without.

To ensure that our $CH_4$ flux estimates were not affected by combustion, measurements of CO were used as an atmospheric tracer for air influenced by biomass burning events or oil development off-gassing, which co-emit CO and $CH_4$. Any profile which had measurements of CO exceeding 150 ppb within the mixed layer were not used in the analysis. This threshold was determined by observing that the annual distribution of CO deviates from normal above 150 ppb (Chang et al., 2014). After CO screening, each individual vertical profile was assessed on a case-by-case basis to ensure that the WRF-STILT model had adequately captured a mixed layer, i.e. the footprint sensitivity attenuated to zero as altitude increased and that the estimated mixed layer height from WRF-STILT clearly separated the mixed layer from the free troposphere. In general, if $h$ defined by WRF-STILT was below 500 m, the profile was rejected due to difficulty in calculating the column enhancement since there was only one binned point below $h$. Of the 273 profiles identified, 9 were rejected due to excessive CO mole fractions, 68 were rejected due to a poor separation of the mixed layer and the free troposphere, and a further 50 were rejected when $h$ differed by more than 750 m using the two methods described in Sect. 3.3. The remaining 146 profiles, just under half of all the





profiles, were kept for analysis. While this is a severe reduction in data, footprint sensitivity plots comparing the total footprint sensitivity of all the receptor points within all profiles against the total footprint sensitivity of those within the profiles kept for analysis showed no significant bias. As a result, this subset is thought to be both representative of the CARVE sampling campaign and free from errors in integration limits.

## 3.5 Eco-region dynamics

The previous assumption that the study region emits $CH_4$ uniformly is correct only to the zeroth-order and serves only as the best estimate for the total magnitude of flux. However, this assumption misses much of the spatiotemporal heterogeneity observed in emissions across and even within different biological and hydrological regimes (as summarized by Olefeldt et al., 2013). At regional scales, such as in this study, it is useful to separate the domain into eco-regions which group regions with similar vegetation, elevation, soil type and hydrologic flow. One can then use the different eco-regions as a basis set of independent sources and sinks in a linear inversion. The eco-regions used within this study were taken from the Environmental Protection Agencies (EPA) Level II map of eco-regions and grouped into the three following land types: tundra, which includes 'Alaska Tundra'; boreal, which includes 'Alaska Boreal Interior', and 'Taiga Cordillera'; and mountains, which includes 'Brooks Range Tundra', 'Marine West Coast Forest', and 'Boreal Cordillera' (Commission for Environmental Cooperation, 1997) (see Fig. 1). While 'Marine West Coast Forests' are not mountainous by definition, a study of $CH_4$ fluxes north of 50°N identified heavily forested areas as being negligible sinks of $CH_4$ (Olefeldt et al., 2013). Therefore these forests are suitably grouped with mountains under the assumption that footprint sensitivity from these areas did not affect observed $\Delta CH_4$.

By calculating the mean fraction of the influence of each eco-region on our measurements using the footprint sensitivity maps, it is possible to estimate the $CH_4$ flux from each eco-region using multiple linear least-squares regression according to the following:

$$\hat{E} = \sum_i f_i E_i, \tag{5}$$

where $f_i$ are the fractions of influence from different eco-regions, $\hat{E}$ are the uniform $CH_4$ flux estimates, and $E_i$ are the $CH_4$ fluxes from each eco-region. These estimates were grouped by month across all three years to maintain a sufficient sample size. To test the assumption that emissions from the mountain land type negligibly affected $\Delta CH_4$, $CH_4$ flux was estimated with and without the mountain land type and presented in Sect. 4.4.2.

## 3.6 Regression analysis

To explore how the variability in $CH_4$ flux estimates is related to ecological, biological and hydrological parameters, the footprint sensitivity functions were also used to calculate weighted means of different variables common in process-based models of $CH_4$ flux. Maps that were included in this analysis were: surface, 10-cm and 40-cm daily soil temperature ($T_x$, where $x$ is the depth [K]) and liquid soil moisture content ($S_x$ [-]) from the North American Regional Reanalysis (NARR)





project; digital elevation ($z$ [m]) from ASTER; days since thaw (DST), derived from passive microwave satellite observations of surface thaw (Steiner et al., 2015); wetlands (%) (Bergamaschi et al., 2007); 30-cm and 100-cm soil organic carbon content ($C_x$, kg/m$^2$) from the Northern Circumpolar Soil Carbon Database (NCSCD) (Hugelius et al., 2013); and percent of soils classed as turbels, histels, gelisols or non-soils from NCSCD (Hugelius et al., 2013).

Previous correlation analyses have suggested that CH$_4$ flux varies non-linearly with variables such as soil temperature (Yvon-Durocher et al., 2014) and elevation (Karion et al., 2016). As such, the following functional forms were used to characterize variability in CH$_4$ flux:

$$E = e^{f(x)} \tag{6}$$

$$f(x) = \begin{cases} \text{E}_a\left(\frac{1}{k\overline{x}} - \frac{1}{kx}\right), & \text{Boltzmann-Arrhenius-Type} \\ \left(\frac{1}{x+\overline{x}}\right), & \text{Inverse-Type} \\ (x - \overline{x}), & \text{Linear-Type} \end{cases} \tag{7}$$

where E is CH$_4$ flux, $x$ is a hypothesized predictor variable, eV is the value of an electron volt in Joules (1.602x10$^{-19}$ J), $k$ is the Boltzmann constant (1.38x10$^{-23}$ J/K), and $\overline{x}$ represents the mean sampled value of $x$. This transformation of units is performed such that when the predictor variable is soil temperature, the fitted parameter optimizes the activation energy (E$_a$) in units of eV (Yvon-Durocher et al., 2014).

     Multi-variable fits were also performed using an equation of the form:

$$\hat{E} = e^{f(Z)\hat{\beta}}, \tag{8}$$

where $\hat{E}$ is an Nx1 vector of modelled CH$_4$ flux estimates, $Z$ is an Nx(M+1) matrix with M columns of predictor variables transformed by $f(x)$ and one column of ones, and $\hat{\beta}$ is an (M+1)x1 vector of estimated parameters. We searched through all possible combinations of functional forms by allowing the M columns of $Z$ to take either the Boltzmann-Arrhenius, inverse, or linear forms written in Eq. 7, leaving the column of ones to fit a constant. The parameter vector $\hat{\beta}$ is then constrained using 20 the Levenberg-Marquardt algorithm for non-linear least-squares regression. Subsets of the data (n<N) were also fit to explore behaviour in limiting spatial and temporal cases. The best fit for each subset was chosen if it minimized the Aikake Information Criterion (AIC) (Burnham et al., 2011).

     Since our estimates represent 5 day averages of CH$_4$ flux (see Sect. 3.1 & 3.4), flux estimates from nearby days can be significantly correlated. To compensate for this correlation, CH$_4$ flux estimates were first averaged into 5-day bins before fitting 25 to reduce issues of autocorrelation and to generate independent estimates of CH$_4$ flux. As with the monthly mean estimates, these were weighted by the total footprint influence. This temporal averaging reduced the total number of independent estimates used in the regression from 146 to 68.



## 4    Results & discussion

In the following sections, we present estimates of regionally averaged monthly $CH_4$ flux and the net total $CH_4$ emitted from our study domain from May–September. A discussion of the source of uncertainties in determining $[CH_4]_0$ will follow. Finally, the set of $CH_4$ fluxes estimated across 2012–2014 will be used to motivate discussions of how sampling different eco-regions

across a dynamic range of soil conditions (e.g. soil temperature, soil moisture, soil type) affected the estimated $CH_4$ flux and budget.

### 4.1    $CH_4$ flux estimates

Using the methods described in Sect. 3.4, monthly mean $CH_4$ fluxes for 2012–2014 were estimated from individual aircraft profiles of the atmosphere (Fig. 4). As will be shown in Sect. 4.4, we found that footprint sensitivity from oceans and mountains

were poorly correlated with observations of $\Delta CH_4$. Therefore the estimated fluxes presented in this subsection only represent $CH_4$ flux from the boreal and tundra eco-regions defined in Sect. 3.5. These estimates ranged from 2–36 mg $CH_4$ m$^{-2}$ d$^{-1}$ (95% C.I.) for individual profiles and showed a distinct seasonal cycle that peaked in late July or early August across all years, consistent with tall and eddy-covariance tower studies (e.g. Karion et al., 2016; Zona et al., 2016; Sweeney et al., 2016). Additionally, they provide observational evidence that region-wide $CH_4$ flux may on average be as high as 5 mg m$^{-2}$ d$^{-1}$ in

the colder months of April and November, also consistent with previous studies on the North Slope (Zona et al., 2016) and interior (Karion et al., 2016) of Alaska.

Monthly mean flux estimates were found to be higher than those inferred from the CRV tower near Fairbanks for the same months (3–9 mg $CH_4$ m$^{-2}$ d$^{-1}$) (Karion et al., 2016), although the ranges from the two studies overlap. As discussed by Karion et al. (2016), the tower observations likely underestimate $CH_4$ flux compared to the aircraft observations because of

the aircraft's increased sensitivity to the North Slope and southwestern Alaska, regions that are known to be seasonal wetlands (and therefore a significant $CH_4$ source (Bergamaschi et al., 2007)), as compared to the interior sites which were more sensitive to upland regions that are thought to emit less $CH_4$ (Olefeldt et al., 2013).

### 4.2    $CH_4$ budget calculations

Integrating over the months sampled every year (May–September), we estimate $CH_4$ emissions from our study region to be

$2.1 \pm 0.5$ Tg, $1.7 \pm 0.4$ Tg and $2.0 \pm 0.3$ Tg for 2012–2014, respectively, with the assumption that all land surfaces (excluding mountains) emit at a uniform rate over the entire month. As our observations do not extend throughout the colder months, we do not provide annual budget estimates since other studies have found that significant emissions of $CH_4$ are observed in the shoulder and cold seasons (Zona et al., 2016; Sweeney et al., 2016; Karion et al., 2016).

Previous estimates of emissions from Alaska during the growing season include the study by Chang et al. (2014), who used

a similar method to estimate the May–September 2012 emissions to be $2.1 \pm 0.5$ Tg, as well as the geostatistical inversion of the CARVE observations by Miller et al. (2016), who estimated May–October $CH_4$ emissions of $1.80 \pm 0.45$, $1.65 \pm 0.43$ and $1.77 \pm 0.45$ Tg $CH_4$ for 2012–2014, respectively. Our mean estimates are within the uncertainties of these other studies,




especially when we account for the $\sim$20% greater area in our study domain. It is promising that our relatively simple method for calculating budgets using selected profiles from the CARVE aircraft observations arrives at similar estimates to values derived from the much more complex geostatistical inverse model used by Miller et al. (2016), particularly as their study was constrained by all the aircraft observations as well as hourly averaged observations from the CRV tower.

Our estimates of May–September $CH_4$ flux, have a mean of $2.0 \pm 0.4$ Tg, and show no significant difference over the three years. These results are consistent with the findings reported by Miller et al. (2016), who suggest that regional $CH_4$ emissions would require decades to respond to changes in surface conditions. Similarly, a recent analysis of long-term measurements of $CH_4$ flux on the North Slope in Alaska observed little change in boundary layer $CH_4$ enhancement over the past 29 years, despite increases in air temperature (Sweeney et al., 2016). This lack of trend could potentially be related to methanogen

community structure in the Arctic as recent microbiological research has found that communities from Arctic soils that were incubated at lower temperatures were insensitive to substrate manipulations, indicating that Arctic methanogens may not be sensitive to the addition of new labile carbon from thawing permafrost (Blake et al., 2015). While it is true that local-scale permafrost degradation patterns such as thermokarsts can result in local $CH_4$ fluxes of >100 mg m$^{-2}$ d$^{-1}$ (Johnston et al., 2014), multi-decadal studies such as Sweeney et al. (2016) suggest that at a regional scale, $CH_4$ fluxes in Alaska have been

stable.

### 4.3   $CH_4$ background estimates

#### 4.3.1   Comparison of background $CH_4$

Since our calculations of $\Delta CH_4$ rely on the free tropospheric $[CH_4]$ to represent background $[CH_4]$ in the mixed layer, it is important to demonstrate that these values are comparable. As seen in Fig. 5, a distinct seasonal cycle in $[CH_4]_0$ is evident and

is consistent with cycles observed in the NOAA Global $CH_4$ Network (Dlugogencky, 2016). This gives us confidence that our calculated $[CH_4]_0$ is not strongly affected by changes in circulation due to long range transport or stratospheric intrusion of clean air.

    To assess the accuracy of our estimates of $[CH_4]_0$, we compare our values to in situ observations of boundary layer $CH_4$ background observed at the Barrow Observatory ground station (BRW: 71.3°N, 156.6°W) in 2013 and 2014 (Dlugogencky,

2016). Because the station did not measure $[CH_4]$ in 2012 (Sweeney et al., 2016), we instead compared the 2012 CARVE $[CH_4]_0$ estimates to the free tropospheric backgrounds measured monthly at the Poker Flats site in interior Alaska (Global Monitoring Division, 2016). We also compared our estimate of $[CH_4]_0$ to the backgrounds determined for the CRV tower (Karion et al., 2016). These other estimates of $CH_4$ background levels were generally within the standard deviation of $[CH_4]_0$ estimated from the CARVE profiles ($\pm 8$ ppb in 2012 and $\pm 6$ ppb in 2013 and 2014). The exceptions were August 2013 and

May 2014 when the CARVE observations were lower than the boundary layer $CH_4$ background observed at BRW, possibly resulting in an overestimation of $CH_4$ flux. It should be noted that the BRW site is located on the northern shore of Alaska and is separated from the Alaskan interior by the Brooks Range so it is not always influenced by the same air mass that affects the remainder of the study region. While the backgrounds estimated from the CRV tower for August 2013 are within our variability,





the May 2014 background estimate is between our estimate and the levels observed at BRW. To evaluate the magnitude of this effect on the estimated May–September budgets, the $CH_4$ flux using the BRW ground station observations were calculated for those two months. Resulting May–September budgets using these new values are 1.5±0.2 and 1.8±0.4 Tg $CH_4$ for 2013 and 2014, respectively. Since these values are within the uncertainties of our initial budget estimates and our background estimates

correspond to those from the CRV tower, we believe that our estimates of $[CH_4]_0$ are representative of background levels in the mixed layer.

### 4.3.2 $CH_4$ background growth rate

Across the entire campaign, the estimated $[CH_4]_0$ had a median and standard deviation of 1880 ± 20 ppb, and a distinct seasonal cycle with higher mole fractions in colder months than in warmer months (Fig. 5). This cycle is chemically driven

and is consistent with observations from global $[CH_4]$ observations (Dlugogencky, 2016). From Fig. 5, it is also evident that background $CH_4$ in the free troposphere rose from 2012–2014.

We estimate the atmospheric growth rate using monthly mean CARVE observed $[CH_4]_0$ fit to the function $y = \sum_{k=0}^{1} \sin(2\pi t + \frac{k\pi}{2}) + \sum_{k=0}^{1} t^k$, where time ($t$) is in units of years. This function is a simplified form of the function used by NOAA to estimate the global growth rate of $CH_4$ from their observation network (ESRL, 2016). Fitting with the Levenberg-Marquadt algorithm

for non-linear least-squares, we estimate the atmospheric growth rate in the free troposphere over Alaska to be 9 ± 2 ppb/yr (p<0.001) with a coefficient of determination of 0.77. The estimated growth rate is consistent with the 9 ± 1 ppb/yr and 8.6 ± 0.6 ppb/yr observed at Barrow (11 m a.s.l.) and Mauna Loa, HI (3397 m a.s.l.) respectively.

Overall we find that the monthly mean and annual growth rate determined from CARVE was the same as BRW within the variability of our observations, with the exception of a few months. These results indicate that local mixed layer $[CH_4]_0$ in

Alaska can be constrained from free tropospheric measurements and gives us confidence in our estimates of $\Delta CH_4$.

### 4.4 Eco-region Dynamics

### 4.4.1 Tundra & Boreal Eco-regions

Figure 6 shows the monthly emissions estimated for the tundra and boreal eco-regions over all three years estimated using the linear system in Eq. 5. The seasonal average $CH_4$ flux from tundra regions was 21 ± 3 mg m$^{-2}$ d$^{-1}$ and ranged from 6–34

mg m$^{-2}$ d$^{-1}$. Although comparable, our mean is lower and our range is narrower than those reported in a database of flux observations at the chamber-scale compiled by Olefeldt et al. (2013), who found that average $CH_4$ flux from wet tundra north of 50 N was 64.5 mg m$^{-2}$ d$^{-1}$ and ranged between 31.9 and 100.6 mg m$^{-2}$ d$^{-1}$. This is likely a result of our estimation method, which relies on a large degree of spatial and temporal averaging which smooths out the high $CH_4$ bursts that can be captured by flux chambers. Our estimates are also comparable to flux observations in 2013–2014 from eddy-covariance towers

on the North Slope, where the monthly mean emissions from May to September ranged from 6–21 mg m$^{-2}$ d$^{-1}$ (Zona et al., 2016), as well as from the Yukon River Delta (25 mg m$^{-2}$ d$^{-1}$) (Fan et al., 1992).





The mean monthly flux was much lower from boreal regions. On average it was $10 \pm 2$ mg m$^{-2}$ d$^{-1}$, and ranged from $4 - 21$ mg m$^{-2}$ d$^{-1}$. The database by Olefeldt et al. (2013) variously classified areas within the boreal region as bogs, fens and palsas, which emit on average $7 - 37$ mg CH$_4$ m$^{-2}$ d$^{-1}$ (Olefeldt et al., 2013). An additional study, which made flux measurements using an eddy covariance tower within a poorly drained forested region in Alaska, found that CH$_4$ flux varied from $3 - 11$ mg m$^{-2}$ d$^{-1}$ in the wettest regions (Iwata et al., 2015). Again, as with the tundra ecosystem, our spatial and temporal averaging method will smooth any CH$_4$ bursts.

Considering the period of May–September, the total flux from both ecosystems ($2.2 \pm 0.2$ Tg) was consistent with the mean of the net budget calculated in the previous section assuming the ecosystems were identical. This analysis also estimates that the tundra eco-region, which represented 18% of the total study area, accounted for more than half of the total flux. Nevertheless, emissions from boreal regions cannot be neglected in estimates of the regional budget since their spatial coverage is quite extensive.

In Fig. 6, emissions from boreal regions appear to lag tundra regions by one month. This offset could be due to a more rapid onset of spring thaw in the tundra eco-region, with maps of freeze-thaw state showing that this area thawed 4–5 days earlier than the boreal eco-region in 2013 and 2014 (Steiner et al., 2015). While this is less than 1 week, a study relating the date of thaw to the annual radiation budget estimated that a 4-day shift in freeze/thaw date alters the annual radiation budget by 250 MJ m$^{-2}$, significantly altering the early season budget (Stone et al., 2002). However, in 2012, the boreal eco-region actually thawed 9 days earlier than the tundra. Considering that fluxes calculated from all years are included in this regression, it is not clear that the difference in seasonal pattern could be explained by differences in thaw date alone. Instead, the difference in the seasonal cycle is more likely a result of the distribution of wetlands. Using a map of wetland extent, it is estimated that 30% of the tundra eco-region can be classified as a seasonal wetland, while only 15% of the boreal eco-region is similarly classified (Bergamaschi et al., 2007). As wetlands are defined by a near-surface water table, methanogenesis can begin when the depth of thaw is much shallower, resulting in a CH$_4$ emissions that can begin much earlier. In Sect. 4.5, the effect of the water table on the relationship between temperature and CH$_4$ flux will be discussed further.

### 4.4.2 Mountain & ocean eco-regions

To test our assumption that emissions from oceans and mountains did not significantly influence measured [CH$_4$], we calculated fluxes assuming any surface (ocean or terrestrial) could emit CH$_4$. The same eco-region regression was used but included the fraction of the footprint which covered the tundra, boreal, mountain, and ocean regions.

Oceans were a weak CH$_4$ source, emitting an average $2 \pm 1$ mg m$^{-2}$ d$^{-1}$ across the study months. However, Student t-testing found that the fraction of oceans sampled had a consistently large p-value across the season ($p > 0.05$), suggesting that footprint sensitivity in oceans was not correlated with $\Delta$CH$_4$. This is consistent with a recent study of summertime sea-air flux of CH$_4$ around Svalbard, Norway, which measured low boundary layer CH$_4$ enhancements despite substantial surface ocean concentrations of CH$_4$ from subsea clathrate deposits in the high arctic (Myhre et al., 2016).

This regression also indicated that mountains might act as a weak seasonal sink of CH$_4$, with a mean strength of $-1 \pm 2$ mg m$^{-2}$ d$^{-1}$, consistent with recent regional observations of CH$_4$ in the mineral soils of Greenland (Jørgensen et al., 2015).



However, the statistics (p>0.1) were too weak to confirm this as a regional scale phenomenon in our study region. In light of these results, the oceans and mountains were masked from $CH_4$ flux estimates, as their inclusion would have diluted the contributions of other land types.

### 4.5    Temperature dependence of $CH_4$ flux

The results of the regression analysis described in Sect. 3.6 are listed in Table 1 and indicate the $r^2$ of the best model, the driving variables, and the functional type for each subset (see Eq. 7). Overall, $CH_4$ flux was seen to most strongly correlate to variance in a Boltzmann-Arrhenius function of $T_{40}$ and the inverse of elevation. While these relationships have been observed in our study region in the past, primarily at the smaller scales of chambers and eddy-covariance towers (e.g. Olefeldt et al., 2013; Sturtevant et al., 2012; Zona et al., 2016), far fewer studies have reported them at a regional scale (Miller et al., 2016; Karion et al., 2016). These latter analyses also identified the inverse elevation (Karion et al., 2016) and sub-surface temperatures (Miller et al., 2016) to be important in explaining the variance of observed $CH_4$. As in the work of Karion et al. (2016), we used a relatively simple analysis method to arrive at similar explanatory variables as the more sophisticated geostatistical inversion model used by Miller et al. (2016) but with only a subset of the data. Although the $r^2$ from our analysis may seem low, they are comparable to values derived from regression analysis conducted for chamber studies (Olefeldt et al., 2013). Our results suggest that soil conditions that affect $CH_4$ flux at the more local scale are also relevant at regional scales.

Since both $CH_4$ production and flux can be much stronger in wetlands, (e.g. Olefeldt et al., 2013; Sturtevant et al., 2012), the profiles were subdivided into two groups based on the sensitivity of the profiles to wetlands in the domain. When a profile's footprint sensitivity was ≥20% wetland by area, the profile was categorized as 'Wetland Present'. Conversely, profiles which had sensitivities ≤10% to wetlands by area were categorized as 'Wetland Absent'. The percent of wetland per area sampled by a profile was calculated by averaging a map of wetland fraction (Bergamaschi et al., 2007) weighted by the footprint sensitivity. It was found that for profiles in the Wetlands Absent category, $T_{40}$ and $S_{40}$ were the key predictors in understanding the variability in $CH_4$ flux, suggesting that $CH_4$ was being formed lower in the soil column. By contrast, for profiles in the Wetlands Present category, increases in $CH_4$ flux were correlated with increases in $T_{10}$ and $S_{10}$.

These two regressions highlight the importance of understanding the depth at which methanogenesis (and methanotrophy) occurs. For instance, in non-wetlands, the water table is deeper than in wetlands. As a result, the onset of $CH_4$ production in non-wetland regions can significantly lag wetland areas, as the lower section of the soil column will take much longer to thaw. In fact, maps of soil temperature estimate that, on average, soil at 10-cm in the Wetland Present region thawed nearly 3 weeks before soil at 40-cm in the Wetland Absent regions. In addition, soils at these depths and in these regions warmed very differently: $T_{10}$ in the Wetland Present regions increased at an average rate of 0.10 K day$^{-1}$ from point of thaw to the point of annual maximum, while $T_{40}$ in the Wetland Absent regions increased at only 0.06 K day$^{-1}$. While $CH_4$ production may occur at any depth within an inundated soil column, $CH_4$ produced at lower soil depths can be transport limited (depending on bubble formation or aerenchyma); it is, therefore, intuitive that $CH_4$ produced in wetlands will be more directly correlated to differences in temperature near the surface. Similarly, the dependence of $CH_4$ production (and ultimately, $CH_4$ flux) on soil





moisture will be most pronounced just beneath the water table, where soil moisture is more susceptible to variability than at lower depths.

These dynamics can be extended to explain the seasonal cycles presented in both Figs. 4 & 6. First, as remarked in Sect. 4.4.1, the tundra eco-region possesses more wetlands than the boreal. As a result, the delayed thaw of the 40-cm soil in combination
with slower warming likely explains the delay in boreal eco-region $CH_4$ flux. Second, while maps of footprint sensitivity across all years showed relatively consistent sensitivity to the tundra in the west and southwest of Alaska (see Figs. 1 & 2), they showed that our measurements were most sensitive to the North Slope of Alaska in 2012, relative to the boreal eco-region. On average, ~30%, 20%, and 20% of the footprint sensitivity was from regions classified as tundra in 2012 – 2014, respectively. Since the tundra region possesses more seasonal wetlands, it is therefore not surprising that the 2012 seasonal
cycle of $CH_4$ flux in Fig. 4 is closer to the tundra eco-region in Fig. 6, while the 2013 and 2014 seasonal cycles more closely resemble that of the boreal eco-region. However, as the tundra only represents 18% of the total area of interest it is likely that the sampling in 2013 and 2014 was representative of the eco-regions with respect to their total area, while the sampling in 2012 may have been over-sensitive to the tundra.

Overall, sub-surface soil temperature was seen to be the single best explanatory variable throughout the regression analysis.
Following the work of Zona et al. (2016), the seasonal cycles of $CH_4$ flux are plotted versus soil temperature and coloured by days since thaw in Fig. 7. In this figure, $CH_4$ flux estimates from individual profiles are shown in black points. To highlight the average seasonal trend, these individual estimates were block-averaged into 5-day bins of days since thaw based on satellite retrievals of thaw state (coloured points), before being smoothed by a lowess-filter which locally averaged 35%, or 80-days, of the seasonal cycle (coloured line) (Steiner et al., 2015). Flux estimates from May 2014 were suspected of being over-estimated
and were thus excluded from this plot and the regression that follows.

Of the three depths presented in Fig. 7, the seasonal cycle of $CH_4$ flux has the most consistent, monotonic relationship with $T_{40}$ throughout the study, while the other depths, especially the individual profiles (small black points) show more scatter. We conjecture that the monotonicity of the relationship between $T_{40}$ and $CH_4$ we observe for our study region is reflective of the fact that $CH_4$ production is, on average, taking place near this soil level. Evidence for the production of $CH_4$ at soil depths well
below the surface have been reported at the eddy-covariance scale before. In particular, the counterclockwise hysteresis loop in our observations of $T_{10}$ and $CH_4$ flux is very similar to the relationship observed by eddy-covariance towers at Ivotuk between $CH_4$ and $T_{15}$ (Zona et al., 2016). Wheras observations of $CH_4$ flux in wetland sites exhibited a clockwise "hysteresis" loop, the Ivotuk site itself was much drier than other eddy-covariance sites compared in the study, leading the authors to conclude that the direction of the loop was related to whether $CH_4$ production was occuring above (clockwise) or below (counterclockwise)
the soil depth at which temperature was measured (Zona et al., 2016). Similarly, an eddy-covariance tower near Fairbanks, AK, also observed a counterclockwise hysteresis loop when $CH_4$ flux was plotted against $T_{20}$ (Iwata et al., 2015). It is therefore reasonable to believe that $CH_4$ production is occuring at depths of ~40-cm in the broader Wetland Absent regions as a result of a lower water table. In particular, these drier regions play a large role in the late season (September – December) budget, as the production of $CH_4$ at soil depths well beneath the surface enables $CH_4$ flux to continue even after the surface soil has begun
to freeze (Zona et al., 2016). In the specific case of Ivotuk, $CH_4$ emitted after the surface had frozen represented nearly 30%



of the total annual budget (Zona et al., 2016). Since "Wetland Absent" regions make up 50% of the surface area of the tundra and boreal eco-regions it is important that these regions are not overlooked when modelling $CH_4$ emissions at a regional scale.

From Fig. 7, we fit the mean $CH_4$ flux and $T_{40}$ from the 5-day means to a Boltzmann-Arrhenius equation and determined an activation energy of $0.75 \pm 0.20$ eV. This value is slightly lower than but still within uncertainties of the global mean activation
energy of 0.96 eV (0.86 – 1.07 eV, 95% C.I.) calculated using $CH_4$ flux measurements from static chambers (Yvon-Durocher et al., 2014). Using $T_{40}$ from NARR in this parameterization, we estimate May–September emissions from our study region to be $2.1 \pm 0.3$ Tg, $1.8 \pm 0.3$ Tg and $2.0 \pm 0.3$ Tg for 2012–2014, respectively (only integrating over non-mountainous areas). This simple model does a remarkable job of capturing the growing season budget estimated from the aircraft observations as well as the timing of the peak in $CH_4$ emissions. As more winter time measurements would be necessary to properly constrain
cold season fluxes, estimates of total annual budgets based on this model are not reported.

## 5  Conclusions

Analysis of $CH_4$ column enhancements supplemented by simulated atmospheric transport allowed us to estimate the monthly mean $CH_4$ fluxes from our study domain (50–75 N, 130–170 W). We estimate that domain averaged $CH_4$ flux ranged from 2.0–36 mg m$^{-2}$ d$^{-1}$ and that $2.1 \pm 0.5$ Tg, $1.7 \pm 0.4$ Tg, and $2.0 \pm 0.3$ Tg $CH_4$ were emitted from our domain for 2012–2014,
respectively. These estimates were consistent with more complex statistical methods, indicating that this relatively simple analytical technique, with only a subset of the data, is sufficient for determining regional scale $CH_4$ emissions. The methodology and analysis that followed are therefore useful guidelines for regional monitoring programs which suggest that short, regular profiling of different eco-regions supplemented by fine-scale meteorological modelling can be sufficient to characterize the regional dynamics of the carbon cycle.

Despite the lack of spatial resolution within the $CH_4$ flux estimates, we were able to leverage the atmospheric transport model to inform some basic regression models on how $CH_4$ flux co-varied with different soil variables and characteristics. We found that when we sampled wetlands, $CH_4$ flux co-varied most significantly with $T_{10}$. Conversely, when wetlands were absent, $CH_4$ flux co-varied with $T_{40}$. These two results are consistent with observations of how the water table affects the anaerobic production of $CH_4$ at small spatial scales and emphasize that it is a relevant control at regional scales. Across our
study region, we were able to reasonably predict the May–September $CH_4$ budget using a Boltzmann-Arrhenius model relating $CH_4$ flux to $T_{40}$.

Overall, these regressions provide insight in the differences in seasonal cycles observed across the years and eco-regions. Methanogenesis in wetlands (like the tundra) occur closer to the surface since the water table depth is higher. By contrast, methanogenesis occurs lower in the soil column in regions with less wetlands (i.e. boreal regions). Since surface soils will
thaw earlier in the season than deeper soils, $CH_4$ production begins earlier in wetland regions and ends later in drier regions. As a result, campaigns like CARVE, who sample across eco-regions need to be cautious to evenly sample regions with different subsurface hydrology. Overall our results show that factors found to affect $CH_4$ emissions at scales of 1 m to 1 km are still relevant at the regional scale, suggesting that regional emissions can scale up local scale studies.



Finally, while meteorological differences affected the springtime freeze-thaw transition and fluxes, there was a distinct lack of interannual variability in the overall May–September budget. While observations of $CH_4$ of flux in northern Alasks have proven to be stable at the scale of eddy covariance towers, this study provides additional regional scale evidence that biogenic $CH_4$ fluxes in the domain of study are stable at the multi-year time scale.

5   *Acknowledgements.*  We thank the pilots, flight crews, and NASA Airborne Science staff from the Wallops Flight Facility for enabling the CARVE Science flights. We acknowledge funding from the National Oceanic and Atmospheric Administration and Natural Sciences and Engineering Research Council of Canada (postdoctoral fellowship to R.Y.-W.C.). Computing resources for this work were provided by the NASA High-End Computing Program through the NASA Advanced Supercomputing Division at the Ames Research Center as well as ACENET, the regional advanced research computing consortium for universities in Atlantic Canada. ACENET is funded by the Canada
10  Foundation for Innovation, the Atlantic Canada Opportunities Agency, and the provinces of Newfoundland & Labrador, Nova Scotia, and New Brunswick. Additional thanks to A. Karion, B. Daube, J. Budney, A. Dayalu, E. Gottlieb, M. Pender, J. Pittman, J. Samra, T. Duck and C. Perro for their help. The research described in this paper was performed as part of CARVE, a NASA Earth Ventures investigation.



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



**Table 1.** Correlation coefficients and predictors of $CH_4$ emissions for various linear regression models

| N | Subset Condition | $r^2$ | Predictors | Type |
|---|---|---|---|---|
| | | All | | |
| 68 | – | 0.36 | $T_{40}$, Z | Boltz.-Arrh., Inv. |
| | | Sub-sets | | |
| 27 | Wetlands Present | 0.40 | $T_{10}$, $S_{10}$ | Boltz.-Arrh., Lin. |
| 28 | Wetlands Absent | 0.48 | $T_{40}$, $S_{40}$ | Boltz.-Arrh., Lin. |

$T_x$: mean sampled x-cm subsoil temperature from NARR (K); $S_x$: mean sampled x-cm subsoil liquid moisture fraction from NARR (-); Z: soil surface elevation above sea-level (km).

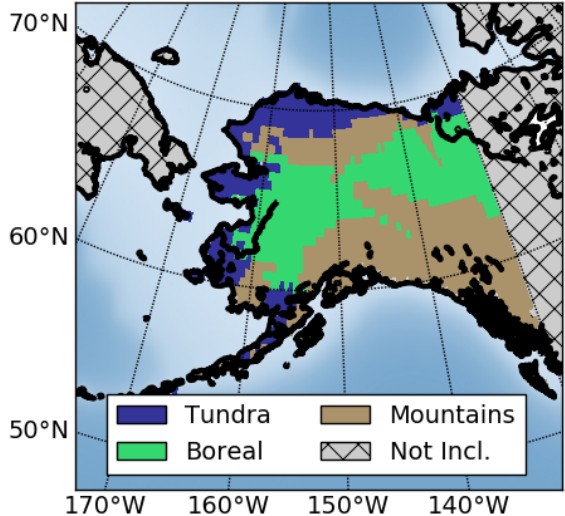

**Figure 1.** Eco-regions derived from the Commission for Environmental Cooperation Level II Terrestrial Eco-regions. The study region is defined by the coastline and filled eco-regions.





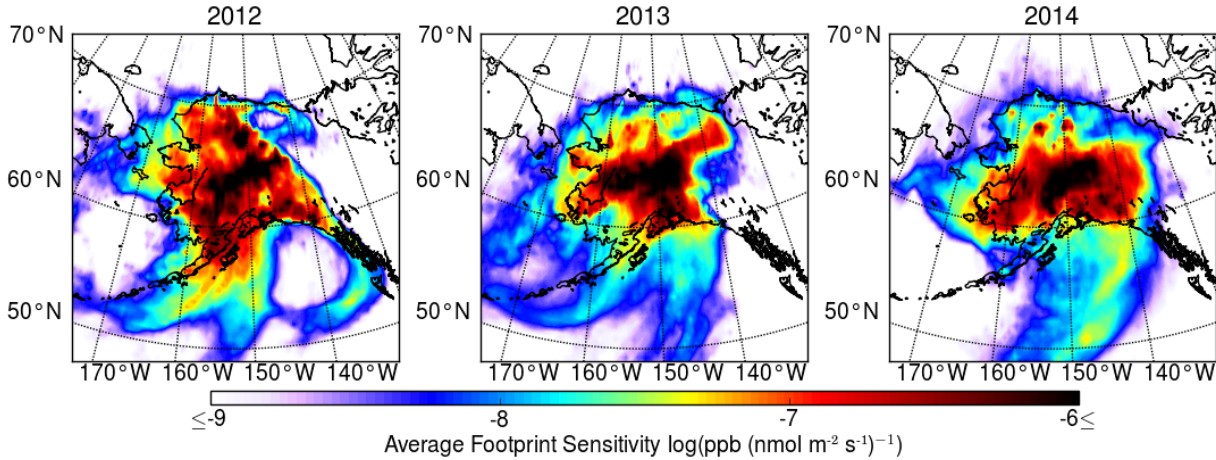

**Figure 2.** The average footprint sensitivity calculated from WRF-STILT is shown for all receptor points modelled within the profiles included in the analysis (2012–2014).

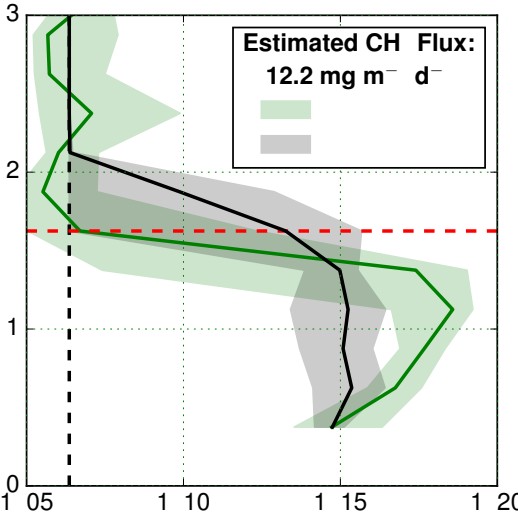

**Figure 3.** A sample profile from 3 Sep 2014 at 65°N, 148.6°W. Shaded regions denote the minimum/maximum of the ranges and the solid line is the median. The dashed red line represents $h$, and the dashed black line is $[CH_4]_0$.



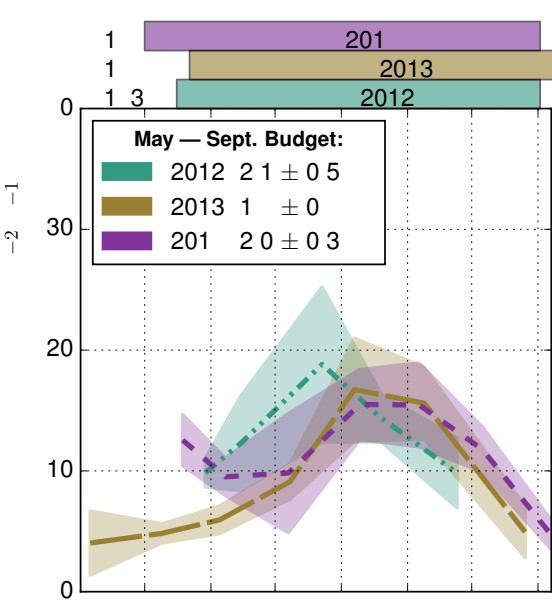

**Figure 4.** Monthly mean $CH_4$ flux estimates. Values are centered on the mean measurement date for a given month and the shaded regions are the standard error of the mean. The bar plot above the graph marks the days when average soil temperatures from NARR were above zero, with the total number of unthawed days given by $n$.




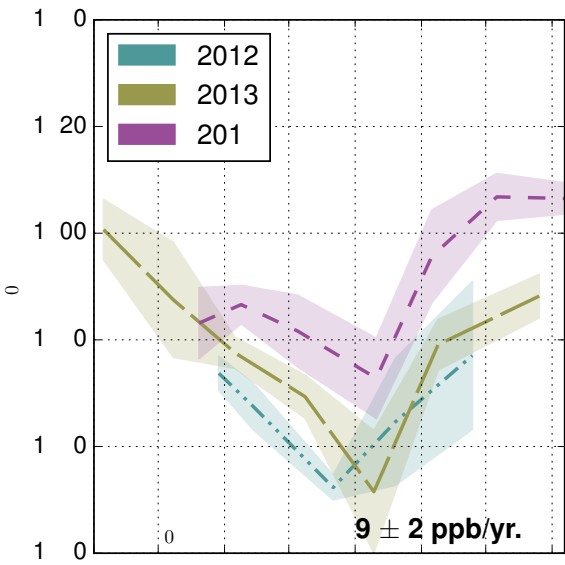

**Figure 5.** The mean tropospheric $[CH_4]_0$ estimated from profiles (shaded areas indicate $1\sigma$). Values are centered on the mean measurement date.

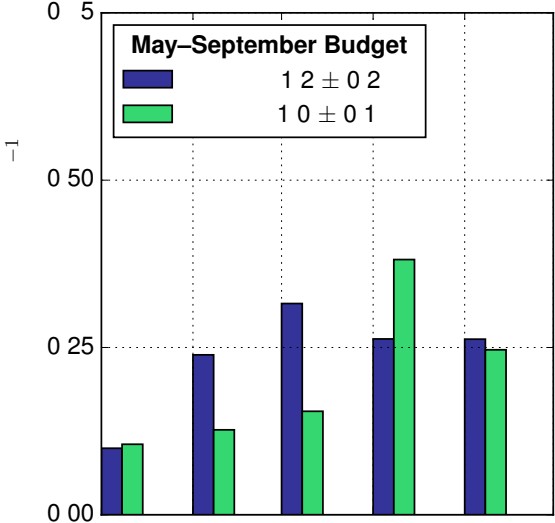

**Figure 6.** Estimated $CH_4$ emissions from tundra and boreal eco-regions averaged over all years.





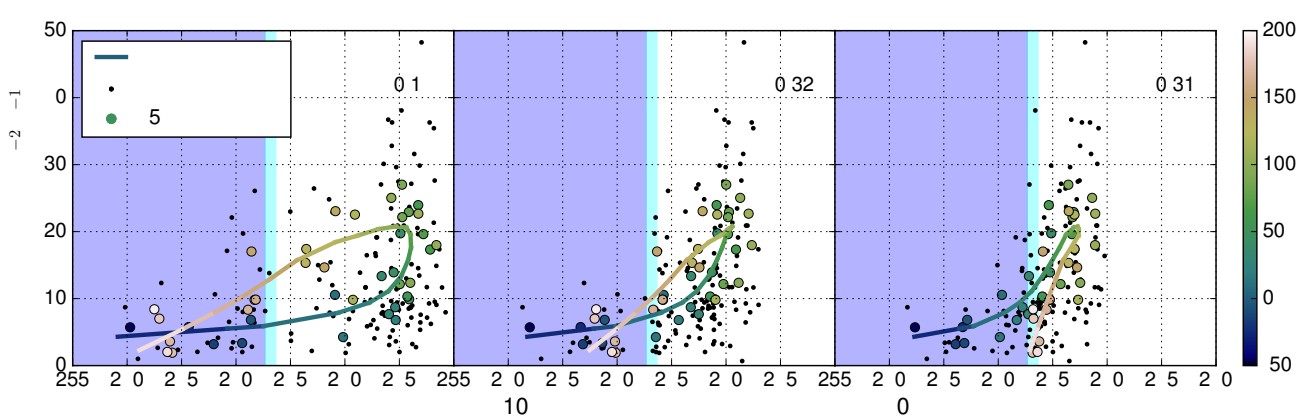

**Figure 7.** (a – c) $CH_4$ flux estimates from atmospheric profiles are shown versus footprint-weighted mean soil temperatures at different depths. The shaded background denotes when the soil temperature was at (cyan) or below (blue) the fusion point of water.