# Peer review of "Estimating regional-scale methane flux and budgets using CARVE aircraft measurements over Alaska"

_Atmospheric Chemistry and Physics, 2017_

## Referee Comment (RC1) · Anonymous Referee #2 · 7 Mar 2017

Hartery et al. presents CH4 fluxes from the Alaskan wetlands derived from aircraft measurements. The paper examines the relationship between the fluxes and several variables. The paper uses an extensive dataset, which has been presented by several previous studies. However, it is interesting to compare different techniques/models to derive fluxes. Examining the drivers of the CH4 flux at such a large scale is novel. The paper is interesting and well written with relatively few typos. The paper should be suitable for publication, but I do have several queries first.

My main concern is that the manuscript is missing a detailed description of how the uncertainties associated with the fluxes were calculated? How were the uncertainties for the individual components estimated and propagated.

I would expect that the choice of background would have a large impact on the calculated fluxes. However, I am not sure how appropriate the CH4 above the mixed layer is for the background. You are assuming that it is representative of air 5 days upwind of the measurements. Wouldn't you expect significant exchange between the PBL and free troposphere over a 5-day period? Is it possible that the free troposphere and mixed layer air could have different histories (e.g. due to long range transport)? You compare your background to those from Karion et al. (2016) who use a "pacific curtain", however I imagine this will only be valid if there is a west to east airflow. Also is Poker Flats in interior Alaska really observing background air? The choice of 5 day sensitivity footprints seems a bit arbitrary, does changing the length of time used to derive the footprints e.g. 5 to 10 days have much impact on the calculated fluxes? Would you use the same background if you used a 10 day sensitivity footprint in your flux calculation?

Figures 3 to 7 appear to have a lot of problems with missing units, axis labels, legends, etc. It looks like the figures have been corrupted. I don't know if this is just a problem for the pdf reader I am using, but please check them. This didn't seem to be a problem with the initial submission. But it does make reviewing some parts of this current version difficult (particularly sections 4.4 and 4.5).

Specific comments

Page 1 line 1. "emissions from northern regions is still poorly constrained". Change "is" to "are".

Page 1, line 10. Change "flux" to "fluxes".

Page 4, line 9 to 11. I find this sentence a bit confusing. I presume you interpolate the instrument's calibration curves between calibration times. I am not sure what the additional interpolation is?

Page 5, line 15. "among other methodological improvements" is a bit vague. Either give more detail about these changes or remove.

Page 5, line 31. You refer to a 5 day footprints here, but on page5, line 17 you say that

footprints were calculated over 10 days. Have I misunderstood something?

Page 11, line 29. For 2012, are there any differences between this study and Chang et al., (2014)? They appear to use identical data and methods.

Page 11, line 29. Can you also compare your fluxes with those from Karion et al (2016)?

Page 12, line 27. Please be more specific about how the background for the CRV tower was calculated.

Page 12, line 20. How large was the difference between CARVE and BRW? Perhaps you could show a scatter plot showing the different methods used to derive the background. This comparison is a bit vague at the moment.

Page 14, line 20. It is worth noting that wetland maps can show significant difference (e.g. Melton et al., Biogeosciences, 2013). I wonder if this would impact on your results in section 4?

Page 15, lines 1-3. What do you mean by "diluted the contributions of other land types"? Does this increase the flux from other land types?

Page 15, line 7. Add correlation coefficients to main text.

Page 16, line 20. Why was May 2014 suspected of being an overestimate?

Page 17, line 33. Suggest you reword e.g. "... regional emissions can be determined by up-scaling local scale studies"

Page 18, line 1-5. Is 3 years really long enough to comment on a lack of inter-annual variability?

Figure 3. The legend doesn't explain what the green and grey shading are, the units in the legend are missing the exponent, missing axis labels. Check formatting.

Figures 4 to 7. These appear to have similar problems to Fig 3. Please check.

---

## Author Comment (AC1) · 7 Mar 2017

[Figure]

**Figure 2.** The average footprint sensitivity calculated from WRF-STILT is shown for all receptor points modelled within the profiles included in the analysis (2012–2014).

[Figure]

**Figure 3.** A sample profile from 3 Sep 2014 at 65°N, 148.6°W. Shaded regions denote the minimum/maximum of the ranges and the solid line is the median. The dashed red line represents $h$, and the dashed black line is $[CH_4]_0$.

[Figure]

**Figure 4.** Monthly mean $CH_4$ flux estimates. Values are centered on the mean measurement date for a given month and the shaded regions are the standard error of the mean. The bar plot above the graph marks the days when average soil temperatures from NARR were above zero, with the total number of unthawed days given by $n$.

[Figure]

**Figure 5.** The mean tropospheric $[CH_4]_0$ estimated from profiles (shaded areas indicate $1\sigma$). Values are centered on the mean measurement date.

[Figure]

**Figure 6.** Estimated $CH_4$ emissions from tundra and boreal eco-regions averaged over all years.

[Figure]

**Figure 7.** (a – c) $CH_4$ flux estimates from atmospheric profiles are shown versus footprint-weighted mean soil temperatures at different depths. The shaded background denotes when the soil temperature was at (cyan) or below (blue) the fusion point of water.

---

## Referee Comment (RC2) · Anonymous Referee #1 · 28 Mar 2017

**Review of Hartery et al: Estimating regional scale methane flux and budgets using CARVE aircraft measurements over Alaska**

**Summary:**

This paper uses aircraft observations of in situ trace gas concentrations and thermodynamics to constrain Lagrangian-transport-inversions of CH4 flux during a campaign over the Alaskan region as part of the CARVE project. More specifically, spatially resolved fluxes of biogenic CH4 flux during the growing seasons between 2012-2014 have been derived. The work uses a rich measurement dataset that has been carefully obtained and calibrated to a high standard.

After a description of the measured dataset, the work goes on to describe a flux derivation method using footprint sensitivities and inversions calculated using WRF-STILT. Printer-friendly version

These methods are interesting and adapt existing conventional optimal flux transportinversion approaches to attempt to link spatially-resolved flux to soil temperature and depth as a function of time. Such an attempt is highly challenging and this paper is a trailblazer in terms of attempting to do this from a long-term aircraft campaign.

That said, I do have some major questions and concerns about the methods and the conclusions drawn from them. At the moment, the flux work seems to be predicated on assumption after assumption, followed by a procedure of arbitrarily discarding data (over half of it in the end), temporal and spatial averaging, and averaging some more, and then discarding more data, before using only 68 (or is it 146?) measured aircraft profiles (averaged to mixed-layer partial columns) to obtain biogenic growing season flux as monthly-averages over 3 years. This equates to about 4.5 profiles per month (though this is only a rough average calculation as there is no information on the sampling statistics per month other than the total number of profiles used across the whole study). I find it hard to accept that such a limited dataset can derive robust flux statistics representative of regional monthly means, especially where the only uncertainty given on the fluxes is the standard error on the mean of the already monthly-averaged fluxes. Such an error statistic is useless - it neither represents the systematic error associated with the method, nor represents the natural variability of CH4 flux in the region between each independent flux calculation. Instead, it convolves the two with no possible determination of which dominates.

What I instead believe the authors have here are a set of independent flux retrievals (one per profile/flight) and independent posterior flux uncertainties using their method. The extensive averaging of these independent retrievals in the paper make it very difficult to assess the performance of the method; and the current error budget is meaningless. Until I can see more about the performance of the method and the statistics of flux retrieval-by-retrieval, I don't have any confidence in the conclusions and discussion later on (e.g. on soil temperature relationships).

The paper does present some very interesting analysis and I believe there is some
really great science to come from the work. Therefore, I definitely recommend publication in ACP as the methane flux problem is a key topic in atmospheric and geoscience at the moment and this paper represents an exciting way to make use of long-term aircraft datasets. However, I do have to recommend major revisions at present as I think the way the analysis has been done needs to be extensively rewired to present more meaningful data that the reader can more transparently assess, especially with regard to independent flux calculations and uncertainty and error budgets. I'll try to give some specific constructive guidance on this below, which I hope would help will turn a questionable analysis into something really quite interesting and useful. My review won't discuss the flux-soil relationships as until I can see the results from the revisions below I don't feel I have enough information to assess the later aspects of the paper.

**Specific comments:**

**1. Abstract, line 9 etc:** It is very important to note that the whole analysis of the paper derives "net emissions", not "emissions", e.g. the statement that "....Boreal emissions....accounted for the remainder of the emissions" should contain the word "net" as the study does not address local or regional sinks (albeit potentially small). This important point needs to be kept in mind throughout the paper when discussing flux and needs to be very clear to the reader early on.

**2. P.2. line 31-32:** I agree that scaling local fluxes to regions is challenging, even in areas where it may be argued it is possible to derive meaningful regional statistical parameterizations (such as this paper sets out to do). However, there are some studies (not currently cited) that have attempted to do this (also at high latitudes) using a combination of aircraft, chamber and eddy covariance measurements. It would be useful to discuss and cite such work in this paper (as it seems very relevant to the introduction, and later discussion, in this paper). Please see: O'Shea, S. J et al.: Methane and carbon dioxide fluxes and their regional scalability for the European Arctic wetlands during the MAMM project in summer 2012, Atmos. Chem. Phys., 14, 13159-13174, doi:10.5194/acp-14-13159-2014, 2014.

ACPD
3. P.5: Footprint method: The authors have used WRF-STILT to derive a surface sensitivity footprint. The footprint seems to have been derived using a grid with frequency/counts equal to the summed incidences of residence of 500 reverse-Lagrangian particles per measurement in the "lower half" of the boundary layer. This is a can of worms and it is glossed over far too quickly here, and later on. I guess this definition is a bit arbitrary and I have plenty of sympathy with it, as defining surface influence in Lagrangian trajectories is very difficult to guantify. However, I would have liked to have seen more discussion on the uncertainty or sensitivity that this arbitrary definition of surface contact may have on the derived flux footprint. The authors at the very least need to clearly acknowledge that there may be an unquantified source of model transport error coupled with the use of their lower half PBL definition; and - more usefully - they could examine footprint sensitivity to different surface contact definitions (e.g. as other percentages of the boundary layer height). I raise this simply because there is no basis to believe that the lower half of the PBL is in dynamical contact with the surface, especially in enormously diurnally-variant boundary layers such as those in Arctic spring. It could be argued that the diurnal ventilation and contraction (i.e. entrainment and detrainment) of the boundary layer could skew the footprint derived here to one more biased toward representing daytime flux (as daytime PBL trajectories that are isentropically detrained into a PBL-residual layer at night-time would not be counted at night-time in the footprint using the authors' method along the 5-day history used). This could have implications for the fluxes that are derived and their biogenic interpretation and quantification. I have no major problem with the use of an arbitrary definition such as this, as it attempts to do the best it can with the information available, but I do think the reader needs to be made more aware of the potential limitations and issues with it. And some of this may be quantifiable with a sensitivity analysis to PBL depth-contact versus footprint. Without this, I would have some outstanding questions about the numerical validity of the later flux calculations and what they truly represent. I liked the discussion on what the footprints represent more globally on Page 6 but more needs to be added. In summary, I suggest an easy (making it clear as to the limitations)
fix and a bigger effort (sensitivity) fix to help those following the work to make their own informed judgment. Page 9 lines 1-5 seem to suggest that some effort has been made to examine footprint sensitivity to the inclusion (or not) of discarded profiles so perhaps it could be simple and useful to add some of this to the paper to convince the reader that there is no important bias (even if as an Appendix?).

**4.** Related to the comment above, were 500 particles released per singular Picarro measurement, or were 500 particle released per mixed-layer column used in the later inversion? If the latter, were these equally spaced with altitude in the mixed layer? A sentence to make this clear would help.

5. P.6. Section 3.3 - Mixed layers: This relates very closely to the comment above. If mixed layer height (i.e. instantaneous local PBL height plus residual layer height) is used to derive the fluxes later on, how do you differentiate between the true PBL (which is in contact with the local surface) and the residual layer (which may represent the previous day's local PBL ventilation, or advection from a more distant regional source)? This is very confusing. The method states (page 6, line 26) that the authors use the integrated mixed-layer partial column concentration (not the vertically-resolved measurements) to calculate flux in Section 3.4. This must then surely convolve any local emissions (in the true local PBL) and non-local emission (in any residual layer). The authors later go to great lengths to show that any residual layer is not influence by long-range (non-regional) transport but this does not solve the problem of varying airmass histories for the true PBL versus the residual layer when these get clumped into a partial column for the purposes of the inversion. When this singular column concentration is used in STILT and coupled to the footprint described in Section 3.1 (and the issues alluded to in the previous comment), it seems impossible to deconvolve spatially-resolved flux with any true or traceable footprint sensitivity as the column represents an unknown mix of local and non-local surface contact. Again, I have a lot of sympathy (more than it may sound like) with the approach and doing the best job possible with adjoint models. But there is currently no awareness or clarity of these
issues in the text which would alert the reader to the challenges and limitations in the approach. Page 7 goes on to demonstrate that 50

**6. P.8. Using CO as a tracer for combustion CH4 sources:** What about pure fugitive emissions of thermogenic CH4 (where there is no combustion)? This could conceivably lead to an over-estimate of biogenic flux if the remaining profiles contain any significant non-biogenic CH4 from sources not co-emitted with (potentially large fluxes of) CO. I see that only 9 of the profiles were discarded by this definition and the analysis of sensitivity by including them to derive a different flux is useful. This style of analysis starts to give the reader what they need to assess things. With this in mind, I have a few suggestions below to consider:

**Suggested principal corrections:**

**7.** The monthly-averages given may be hiding a wealth of useful data. A time series of biogenic flux (as an area-normalised quanitity – i.e. as biogenic flux per unit time per unit area) for each independent flux retrieval (I believe there are 68 of these?) would be useful. The posterior flux uncertainty (that STILT should yield as output) could be plotted as an error bar on each data point on such a plot.

8. Error/uncertainty analysis: As discussed above, the current tolerance placed on the derived fluxes is meaningless and does not represent either systematic error (flux inversion uncertainty) or natural variability. The seasonal trend plotted in Figures 4 and 5 do not convince me that natural regional variability dominates the mean as this could simply be a manifestation of the changing northern hemispheric seasonal background and priors used. I would suggest that the posterior flux uncertainty of independent retrievals/footprints is used instead as this captures the uncertainty on each retrieval. And then, rather than a standard error on the mean flux (taken from the spread of the averaged inverted fluxes), which would clearly be an incorrect (and much reduced) error, I would recommend quoting the posterior flux uncertainty (calculated as an average of the posterior uncertainties across all inversion that contribute to the final monthly

ACPD
mean). It would be important to give the average of the posterior uncertainties (not their standard deviation or standard error), to yield a meaningful uncertainty on the monthly mean flux. Such an error will still convolve natural flux variability but at least it would be a more accurate measure of the systematic uncertainty in the method used. This should replace the shading (error bars) used in Figure 4.

**9** A table of the derived mean fluxes (and their corrected uncertainties) could be presented, which displays flux with and without the sensitivities/assumptions that have been used (i.e. masking seas and mountains, removing elevated CO profiles). These fluxes are currently in the body of the text, making it hard to compare them. And perhaps a new flux could be calculated where all 248 profiles are used in the inversion? A table would add (at a glance) the comparison between these sensitivities. This latter sensitivity test would then give the reader all the information they need to compare the information and make their own judgement about what they trust and the implications of the assumptions used.

**Technical corrections:**

10. 1/ Title – hyphenate "regional-scale" 11. 2/ Abstract line 1: change to "...gas but its emissions...." Not "their". 12. 3/ P.1. Line 10: change to "...CH4 flux was..." or "...CH4 fluxes were..." - There seems to be some confusion between the use of singular and plural references throughout when referring to flux and fluxes, respectively. Please check as I won't list any further instances. 13. 4/ P.2 line 5: change to "40 °N" and check throughout. "40N" is not acceptable. 14. P.2. Line 9: century should always be capitalized when referring to a specific century. 15. P.4. Line 6: add space to "195 K". 16. P. 8, lime 11: typo - change to "weighting".

---

## Referee Comment (RC3) · Anonymous Referee #1 · 28 Mar 2017

Sorry - latex error with percentages. Sepecific comment 5 is here in full without the hanging sentence....

**5. P.6. Section 3.3 - Mixed layers:** This relates very closely to the comment above. If mixed layer height (i.e. instantaneous local PBL height plus residual layer height) is used to derive the fluxes later on, how do you differentiate between the true PBL (which is in contact with the local surface) and the residual layer (which may represent the previous day's local PBL ventilation, or advection from a more distant regional source)? This is very confusing. The method states (page 6, line 26) that the authors use the integrated mixed-layer partial column concentration (not the vertically-resolved measurements) to calculate flux in Section 3.4. This must then surely convolve any local emissions (in the true local PBL) and non-local emission (in any residual layer). The authors later go to great lengths to show that any residual layer is not influence

by long-range (non-regional) transport but this does not solve the problem of varying airmass histories for the true PBL versus the residual layer when these get clumped into a partial column for the purposes of the inversion. When this singular column concentration is used in STILT and coupled to the footprint described in Section 3.1 (and the issues alluded to in the previous comment), it seems impossible to deconvolve spatially-resolved flux with any true or traceable footprint sensitivity as the column represents an unknown mix of local and non-local surface contact. Again, I have a lot of sympathy (more than it may sound like) with the approach and doing the best job possible with adjoint models. But there is currently no awareness or clarity of these issues in the text which would alert the reader to the challenges and limitations in the approach. Page 7 goes on to demonstrate that 50 percent of profiles were discarded from flux analysis because mixed layers could not be reliably separated or that PBL was "over-estimated" when compared to trace gas profiles. What does over-estimated mean here, and why are the remaining profiles more trustworthy for analysis than those retained? Discarding half of the dataset to select only those data that agree with a contrived and questionable model is a little worrying. I would have liked to know more as to why >50 percent of the dataset is at odds with the model/assumptions and to know what the sensitivity to including the data might be. Can you be confident that the way the remaining 50percent has been retained hasn't led to some systematic bias in the data and model treatment?

---

## Author Comment (AC2) · 1 Jul 2017

**Summary**

Hartery et al. presents CH4 fluxes from the Alaskan wetlands derived from aircraft measurements. The paper examines the relationship between the fluxes and several variables. The paper uses an extensive dataset, which has been presented by several previous studies. However, it is interesting to compare different techniques/models to derive fluxes. Examining the drivers of the CH4 flux at such a large scale is novel. The paper is interesting and well written with relatively few typos. The paper should be suitable for publication, but I do have several queries first.

My main concern is that the manuscript is missing a detailed description of how the uncertainties associated with the fluxes were calculated? How were the uncertainties for the individual components estimated and propagated.

I would expect that the choice of background would have a large impact on the calculated fluxes. However, I am not sure how appropriate the CH4 above the mixed layer is for the background. You are assuming that it is representative of air 5 days upwind of the measurements. Wouldn't you expect significant exchange between the PBL and free troposphere over a 5-day period? Is it possible that the free troposphere and mixed layer air could have different histories (e.g. due to long range transport)? You compare your background to those from Karion et al. (2016) who use 'pacific curtain', however I imagine this will only be valid if there is a west to east airflow. Also is Poker Flats in interior Alaska really observing background air? The choice of 5 day sensitivity footprints seems a bit arbitrary, does changing the length of time used to derive the footprints e.g. 5 to 10 days have much impact on the calculated fluxes? Would you use the same background if you used a 10 day sensitivity footprint in your flux calculation?

Figures 3 to 7 appear to have a lot of problems with missing units, axis labels, legends, etc. It looks like the figures have been corrupted. I don't know if this is just a problem for the pdf reader I am using, but please check them. This didn't seem to be a problem with the initial submission. But it does make reviewing some parts of this current version difficult (particularly sections 4.4 and 4.5).

We thank the reviewer for their thoughtful and detailed comments on the manuscript.

The reviewer has insightfully pointed out the greatest uncertainty in our analysis - identifying background CH4 levels. As the reviewer summarizes, we assume that CH4 levels in the free troposphere are representative of background levels in the mixed layer. As we cited in the submitted manuscript, this is not unusual for aircraft studies (Chang et al., 2014; Gatti et al., 2014; Chou et al., 2002) and has been conducted successfully for the CO2 analysis from this campaign (Commane et al., 2017, now cited in the main text). As evidence that this is a valid assumption, we compared our free tropospheric levels with surface observations at the Barrow site in Alaska. This is illustrated in a new figure now included in the Supplement. As requested by the reviewer, backgrounds determined for the CRV tower are also included in this new figure. As can be seen, the Barrow observatory tends to observe slightly higher mole fractions of CH4, although more variable, than observed in the free troposphere or from the pacific curtain used by Karion et al. One could conclude that the free tropospheric levels are too low, resulting in a higher flux estimate. However, the background CO2 measured at Barrow and the free tropospheric CO2 measured during CARVE, show very little, if any, bias (Commane et al., 2017). To be more transparent about these issues, we have expanded the text in Sect. 4.3 and direct the reader to the new figure in the Supplement so that they can judge our assumptions for themselves.

Another piece of evidence that air in the mixed layer does not mix up into the free troposphere is that vertical profiles of the cumulative 5-day footprint sensitivity determined by WRF-STILT tend towards zero in the free troposphere (e.g. Fig. 3 in the main text). This would suggest that the free troposphere is not being influenced by the surface in our domain, and is therefore not influenced by air from the mixed layer. As the reviewer suggests, however, it is possible that air in the mixed layer and the free troposphere have different transport histories outside of our domain and we now discuss this in Sect. 4.3.1. Unpublished data from measurements taken on the Alaskan Coast Guard aircrafts off the northern coast of Alaska show that average methane profiles are fairly constant from the surface to 7 km, suggesting that free tropospheric levels do represent surface background levels (measurements described by Karion et al., 2013). However,

this may not be true for air transported from other directions. It should be noted that the majority of the air enters our domain from over the ocean, with only approximately 10% originating from the region east of our domain.

To address both reviewers' concerns about backgrounds, we now include a new section (Sect. 3.7) describing how we determine and propagate the uncertainties in our estimates. As the reviewer surmises, the background CH4 mixing ratio contributes the greatest uncertainty to our calculations, now shown in Table S4.

With regards to 10-day footprints, the observed free tropospheric mixing ratio that we use for our background obviously remains constant. The modelled footprint sensitivity does not significantly change if the run time is extended further back because the air will have moved outside of our study domain (50-75°N and 130-170°W) and only surface influences from inside our domain are included in our analysis of the modelled column enhancement. As such, modelled column enhancements from the 5-day cumulative footprints closely resemble those calculated from the 10-day cumulative footprints.

Finally, we apologize for the figures not displaying properly in the Discussions paper. We will ensure that we double check that there are no more compiling issues in the future.

**Specific comments**

**Page 1 line 1. "emissions from northern regions is still poorly constrained". Change 'is' to 'are.'**

This change has been implemented at the reviewer's suggestion.

**Page 1, line 10. Change 'flux' to 'fluxes'.**

This change has been implemented at the reviewer's suggestion.

**Page 4, line 9 to 11. I find this sentence a bit confusing. I presume you interpolate the instrument's calibration curves between calibration times. I am not sure what the additional interpolation is?**

The sentence was meant to convey that we interpolate using the calibration curve determined by the low and high spans. The reference to the second interpolation has been removed and the text now reads "These in-flight calibrations were linearly interpolated between calibration times to generate time-varying calibration curves".

**Page 5, line 15. 'among other methodological improvements' is a bit vague. Either give more detail about these changes or remove.**

This line has been removed as the relevant considerations are as already stated.

**Page 5, line 31. You refer to a 5 day footprints here, but on page 5, line 17 you say that footprints were calculated over 10 days. Have I misunderstood something?**

WRF-STILT footprints were calculated for a total of ten days by Henderson et al. (2015). However, our study only uses the first five days because the air had mostly exited our study domain by then. To clarify this, we now state this clearly in the last

paragraph of Sect. 3.1.

**Page 11, line 29. For 2012, are there any differences between this study and Chang et al., (2014)? They appear to use identical data and methods.**

This study covers a slightly larger east-west domain (130–170°W vs 140–170°W) and only reports fluxes from non-mountainous land. The primary reason for including the 2012 analysis was so that the results could be compared over all three years using a consistent analysis method. It also served as a means of checking our current method.

**Page 11, line 29. Can you also compare your fluxes with those from Karion et al (2016)?**

In the submitted manuscript, we compared our fluxes with those from Karion et al. (2016) in the last paragraph of Sect. 4.1. The section that the reviewer listed is in the budget calculations where we do not compare with the study by Karion et al. because they do not report an estimated budget. It would be inappropriate to extrapolate their results to our entire domain since the CRV tower is not sensitive to the same surface types.

**Page 12, line 27. Please be more specific about how the background for the CRV tower was calculated.**

As discussed by Karion et al. (2016), backgrounds for the CRV tower were calculated by following particle trajectories in WRF-STILT backwards until they crossed a Pacific basin boundary "curtain", which is determined based on an interpolation of observations. These details have now been included in the text.

**Page 12, line 20. How large was the difference between CARVE and BRW? Perhaps you could show a scatter plot showing the different methods used to derive the background. This comparison is a bit vague at the moment.**

As recommended by the reviewer, a figure illustrating background CH4 determined from this study, the CRV tower and BRW observations is now included in the Supplement (Fig. S3). The comparison is quite favorable with the exception of those months already mentioned in the text.

**Page 14, line 20. It is worth noting that wetland maps can show significant difference (e.g. Melton et al., Biogeosciences, 2013). I wonder if this would impact on your results in section 4?**

Since different wetland maps have different spatial distributions, the results are dependent on the map that we use. In earlier iterations of this analysis we did try different maps, including those compared by Melton et al. (2013). However, we decided to only show results from the map by Bergamaschi et al. (2007) because the focus of the study was to quantify the methane budget and not to optimize wetland maps. The study by Miller et al. (2016) compares CH4 emissions estimated using different wetland maps extensively and we now direct the reader to this study in this section of the main text.

**Page 15, lines 1-3. What do you mean by "diluted the contributions of other land types?" Does this increase the flux from other land types?**

'Dilution' was intended to convey the fact that by averaging over a region where no flux is expected, that the estimated monthly flux will be underestimated in areas where the flux is actually occurring. This does not affect budget calculations and allows comparisons with tower and chamber studies more easily. This sentence has been changed to `"as their inclusion would have led to an underestimation of the net` $CH_4$ `flux attributed to non-mountainous land surfaces"`.

**Page 15, line 7. Add correlation coefficients to main text.**

These have now been added to the main text.

**Page 16, line 20. Why was May 2014 suspected of being an overestimate?**

As was discussed in the submitted manuscript on page 12, line 29–31, the flux estimates in May 2014 are possibly overestimated based on comparisons of observed free tropospheric mole fractions of CH4 by the CARVE aircraft being significantly lower than those observed at either the CRV Tower or at Barrow. At the reviewers request, this is now illustrated in Fig. S3 in the Supplement.

**Page 17, line 33. Suggest you reword e.g. "... regional emissions can be determined by up-scaling local scale studies"**

This has been changed to the reviewer's suggestion.

**Page 18, line 1-5. Is 3 years really long enough to comment on a lack of inter-annual variability?**

We agree with the reviewer here and have removed this entire paragraph.

**Figure 3. The legend doesn't explain what the green and grey shading are, the units in the legend are missing the exponent, missing axis labels. Check formatting.**

**Figures 4 to 7. These appear to have similar problems to Fig 3. Please check.**

We apologize that the figures in the discussion paper were not properly formatted. This will be fixed for the final publication.

**References**

Bergamaschi, P., Frankenberg, C., Meirink, J., Krol, M., Dentener, F., Wagner, T., Platt, U., Kaplan, J., Körner, S., Heimann, M., et al.: Satellite chartography of atmospheric methane from SCIAMACHY on board ENVISAT: 2. Evaluation based on inverse model simulations, Journal of Geophysical Research: Atmospheres, 112, 2007.

Chang, R. Y.-W., Miller, C. E., Dinardo, S. J., Karion, A., Sweeney, C., Daube, B. C., Henderson, J. M., Mountain, M. E., Eluszkiewicz, J., Miller, J. B., et al.: Methane emissions from Alaska in 2012 from CARVE airborne observations, Proceedings of the National Academy of Sciences, 111, 16 694–16 699, 2014.

Chou, W. W., Wofsy, S. C., Harriss, R. C., Lin, J. C., Gerbig, C., and Sachse, G. W.: Net fluxes of CO 2 in Amazonia derived from aircraft observations, J. Geophys. Rese., 107, 4614, doi:10.1029/2001JD001295, http://www.agu.org/pubs/crossref/2002/2001JD001295.shtml, 2002.

Commane, R., Lindaas, J., Benmergui, J., Luus, K. A., Chang, R. Y.-W., Daube, B. C., Euskirchen, E. S., Henderson, J. M., Karion, A., Miller, J. B., Miller, S. M., Parazoo, N. C., Randerson, J. T., Sweeney, C., Tans, P., Thoning, K., Veraverbeke, S., Miller, C. E., and

Wofsy, S. C.: Carbon dioxide sources from Alaska driven by increasing early winter respiration from Arctic tundra, Proc. Nat. Acad. Sci., 114, 5361–5366, doi:10.1073/pnas.1618567114, http://www.pnas.org/content/114/21/5361.abstract, 2017.

Gatti, L. V., Gloor, M., Miller, J. B., Doughty, C. E., Malhi, Y., Domingues, L. G., Basso, L. S., Martinewski, a., Correia, C. S. C., Borges, V. F., Freitas, S., Braz, R., Anderson, L. O., Rocha, H., Grace, J., Phillips, O. L., and Lloyd, J.: Drought sensitivity of Amazonian carbon balance revealed by atmospheric measurements., Nature, 506, 76–80, doi:10.1038/nature12957, 2014.

Henderson, J. M., Eluszkiewicz, J., Mountain, M. E., Nehrkorn, T., Chang, R. Y.-W., Karion, A., Miller, J. B., Sweeney, C., Steiner, N., Wofsy, S. C., and Miller, C. E.: Atmospheric transport simulations in support of the Carbon in Arctic Reservoirs Vulnerability Experiment (CARVE), Atmospheric Chemistry and Physics, 15, 4093–4116, doi:doi:10.5194/acp-15-4093-2015, 2015.

Karion, A., Sweeney, C., Wolter, S., Newberger, T., Chen, H., Andrews, a., Kofler, J., Neff, D., and Tans, P.: Long-term greenhouse gas measurements from aircraft, Atmospheric Measurement Techniques, 6, 511–526, doi:10.5194/amt-6-511-2013, 2013.

Karion, A., Sweeney, C., Miller, J. B., Andrews, A. E., Commane, R., Dinardo, S., Henderson, J. M., Lindaas, J., Lin, J. C., Luus, K. A., Newberger, T., Tans, P., Wofsy, S. C., Wolter, S., and Miller, C. E.: Investigating Alaskan methane and carbon dioxide fluxes using measurements from the CARVE tower, Atmos. Chem. Phys., 16, 5383–5398, doi:10.5194/acp-16-5383-2016, http://www.atmos-chem-phys.net/16/5383/2016/, 2016.

Melton, J., Wania, R., Hodson, E., Poulter, B., Ringeval, B., Spahni, R., Bohn, T., Avis, C., Beerling, D., Chen, G., et al.: Present state of global wetland extent and wetland methane modelling: conclusions from a model intercomparison project (WETCHIMP), Biogeosciences, 10, 753–788, 2013.

Miller, S., Miller, C., Commane, R., Chang, R.-W., Dinardo, S., Henderson, J., Karion, A., Lindaas, J., Melton, J., Miller, J., Sweeney, C., Wofsy, S., and Michalak, A.: A multi-year estimate of methane fluxes in Alaska from CARVE atmospheric observations, Global Biogeochemical Cycles, 30, 1441—-1453, 2016.

---

## Author Comment (AC3) · 1 Jul 2017

**1   Summary**

**This paper uses aircraft observations of in situ trace gas concentrations and thermodynamics to constrain Lagrangian-transport-inversions of CH4 flux during a campaign over the Alaskan region as part of the CARVE project. More specifically, spatially resolved fluxes of biogenic CH4 flux during the growing seasons between 2012-2014 have been derived. The work uses a rich measurement dataset that has been carefully obtained and calibrated to a high standard.**

**After a description of the measured dataset, the work goes on to describe a flux**

derivation method using footprint sensitivities and inversions calculated using WRF-STILT. These methods are interesting and adapt existing conventional optimal flux transport inversion approaches to attempt to link spatially-resolved flux to soil temperature and depth as a function of time. Such an attempt is highly challenging and this paper is a trailblazer in terms of attempting to do this from a long-term aircraft campaign.

That said, I do have some major questions and concerns about the methods and the conclusions drawn from them. At the moment, the flux work seems to be predicated on assumption after assumption, followed by a procedure of arbitrarily discarding data (over half of it in the end), temporal and spatial averaging, and averaging some more, and then discarding more data, before using only 68 (or is it 146?) measured aircraft profiles (averaged to mixed-layer partial columns) to obtain biogenic growing season flux as monthly-averages over 3 years. This equates to about 4.5 profiles per month (though this is only a rough average calculation as there is no information on the sampling statistics per month other than the total number of profiles used across the whole study). I find it hard to accept that such a limited dataset can derive robust flux statistics representative of regional monthly means, especially where the only uncertainty given on the fluxes is the standard error on the mean of the already monthly-averaged fluxes. Such an error statistic is useless – it neither represents the systematic error associated with the method, nor represents the natural variability of CH4 flux in the region between each independent flux calculation. Instead, it convolves the two with no possible determination of which dominates.

What I instead believe the authors have here are a set of independent flux retrievals (one per profile/flight) and independent posterior flux uncertainties using their method. The extensive averaging of these independent retrievals in the paper make it very difficult to assess the performance of the method; and the current error budget is meaningless. Until I can see more about the performance

**of the method and the statistics of flux retrieval-by-retrieval, I don't have any confidence in the conclusions and discussion later on (e.g. on soil temperature relationships).**

**The paper does present some very interesting analysis and I believe there is some really great science to come from the work. Therefore, I definitely recommend publication in ACP as the methane flux problem is a key topic in atmospheric and geoscience at the moment and this paper represents an exciting way to make use of long-term aircraft datasets. However, I do have to recommend major revisions at present as I think the way the analysis has been done needs to be extensively rewired to present more meaningful data that the reader can more transparently assess, especially with regard to independent flux calculations and uncertainty and error budgets. I'll try to give some specific constructive guidance on this below, which I hope would help will turn a questionable analysis into something really quite interesting and useful. My review won't discuss the flux-soil relationships as until I can see the results from the revisions below I don't feel I have enough information to assess the later aspects of the paper.**

We thank the reviewer for their very thorough reading of the paper and constructive comments and suggestions. We believe that the assumptions made in our analysis are not as extreme as the reviewer suggests and that our responses below address some of the confusion caused by our initial descriptions. We have added extra text in the manuscript to clarify these points. We agree that quantifying the uncertainties is very challenging. Since our analysis method is not a Bayesian inversion, rather we simply divide our observed column methane enhancements by the modelled column footprint sensitivity to estimate net fluxes, the model does not actually provide a posterior flux uncertainty. However, we now explore the uncertainties associated with our analysis through bootstrapping / Monte Carlo. This is explained in more detail below and is

described in a new section (Sect. 3.7) of the manuscript.

To address the reviewer's suggestions, we have added a Supplement which includes tables listing all the profiles, whether they were included in our analysis, and the reason for exclusion if relevant, as well as monthly means and uncertainties calculated using different methods. It also includes additional figures showing the flux calculated from individual profiles with uncertainties as described above, the resulting residual when compared to the monthly means and background CH4 observed from the aircraft, at the BRW observatory and inferred from the CRV tower.

**Specific comments**

**1. Abstract, line 9 etc: It is very important to note that the whole analysis of the paper derives "net emissions", not "emissions", e.g. the statement that ". . ..Boreal emissions. . ..accounted for the remainder of the emissions" should contain the word "net" as the study does not address local or regional sinks (albeit potentially small). This important point needs to be kept in mind throughout the paper when discussing flux and needs to be very clear to the reader early on.**

The reviewer brings up a very important point and we have changed the wording throughout the text to reflect both that we are only calculating net emissions and that we are assuming that the emissions are originating from biogenic sources.

**2. P.2. line 31-32: I agree that scaling local fluxes to regions is challenging, even in areas where it may be argued it is possible to derive meaningful regional statistical parameterizations (such as this paper sets out to do). However, there**

**are some studies (not currently cited) that have attempted to do this (also at high latitudes) using a combination of aircraft, chamber and eddy covariance measurements. It would be useful to discuss and cite such work in this paper (as it seems very relevant to the introduction, and later discussion, in this paper). Please see: O'Shea, S. J et al.: Methane and carbon dioxide fluxes and their regional scalability for the European Arctic wetlands during the MAMM project in summer 2012, Atmos. Chem. Phys., 14, 13159-13174, doi:10.5194/acp-14-13159-2014, 2014.**

A citation to this study has now been included.

**3. P.5: Footprint method:**

**The authors have used WRF-STILT to derive a surface sensitivity footprint. The footprint seems to have been derived using a grid with frequency/counts equal to the summed incidences of residence of 500 reverse- Lagrangian particles per measurement in the "lower half" of the boundary layer. This is a can of worms and it is glossed over far too quickly here, and later on. I guess this definition is a bit arbitrary and I have plenty of sympathy with it, as defining surface influence in Lagrangian trajectories is very difficult to quantify. However, I would have liked to have seen more discussion on the uncertainty or sensitivity that this arbitrary definition of surface contact may have on the derived flux footprint. The authors at the very least need to clearly acknowledge that there may be an unquantified source of model transport error coupled with the use of their lower half PBL definition; and - more usefully - they could examine footprint sensitivity to different surface contact definitions (e.g. as other percentages of the boundary layer height). I raise this simply because there is no basis to believe that the lower half of the PBL is in dynamical contact with the surface, especially in enormously diurnally-variant boundary layers such as those in Arctic spring.**

**It could be argued that the diurnal ventilation and contraction (i.e. entrainment and detrainment) of the boundary layer could skew the footprint derived here to one more biased toward representing daytime flux (as daytime PBL trajectories that are isentropically detrained into a PBL-residual layer at night-time would not be counted at night-time in the footprint using the authors' method along the 5-day history used). This could have implications for the fluxes that are derived and their biogenic interpretation and quantification. I have no major problem with the use of an arbitrary definition such as this, as it attempts to do the best it can with the information available, but I do think the reader needs to be made more aware of the potential limitations and issues with it.**

**And some of this may be quantifiable with a sensitivity analysis to PBL depth-contact versus footprint. Without this, I would have some outstanding questions about the numerical validity of the later flux calculations and what they truly represent. I liked the discussion on what the footprints represent more globally on Page 6 but more needs to be added. In summary, I suggest an easy (making it clear as to the limitations) fix and a bigger effort (sensitivity) fix to help those following the work to make their own informed judgment. Page 9 lines 1-5 seem to suggest that some effort has been made to examine footprint sensitivity to the inclusion (or not) of discarded profiles so perhaps it could be simple and useful to add some of this to the paper to convince the reader that there is no important bias (even if as an Appendix?).**

The reviewer brings up some very important concerns that face any receptor-oriented atmospheric model. The initial description in the manuscript was rather sparse so we hope that the discussion in response to this and the next two points will provide more clarity. The text in Sects. 3.1 and 3.4 of the main text have been edited to include more details.

In an early publication describing the development of STILT, Gerbig et al. (2003) in-

vestigated the effects of varying the assumed height at which the air is well-mixed. In their sensitivity analysis, they found no significant changes in their results when the fraction was varied from 10–100% of the boundary layer, although lower fractions resulted in less particles influenced by the surface and therefore increased noise in their results. We acknowledge that their findings were for continental North America using assimilated meteorological data, and may not be representative of Alaska, especially in the spring. Nevertheless, these findings suggest that this parameter is not the greatest source of uncertainty in our analysis. A sentence in the main text now directs the reader to the Supplement where this discussion is now included.

Air detrained into the residual layer at night is not influenced by the surface and should therefore not be included in the footprint sensitivity. As the 500 particles are traced backwards from the receptor point, a number will be in the PBL during the day. As detrainment occurs at night, some fraction of those particles will move into the residual layer while the rest remain in the nocturnal boundary layer. Of the particles remaining in the boundary layer, those that are in the lower half are the ones actually influenced by the surface and therefore included in the footprint sensitivity in the evening hours. We acknowledge, however, that defining the boundary layer at night is difficult and now mention it in this section. Nevertheless, we have fairly high confidence in the representativeness of our footprint sensitivities since past CARVE studies using the same WRF-STILT model coupled to a surface $CO_2$ vegetative flux model have successfully predicted atmospheric $CO_2$ mixing ratios in the mixed layer (Commane et al. 2017, Karion et al. 2016). Since $CO_2$ fluxes are bidirectional and highly diurnal, these studies suggest that the transport model represents surface influences during both day and night reasonably well. This is now discussed in Sect. 3.1 and reference is made to these other studies.

**4.   Related to the comment above, were 500 particles released per singular Picarro measurement, or were 500 particle released per mixed-layer column**

used in the later inversion? If the latter, were these equally spaced with altitude in the mixed layer? A sentence to make this clear would help.

As described in the text, the Picarro measurements below 1 km were aggregated into discrete bins every 5 km in the horizontal and 50 m in the vertical and above 1 km they were aggregated every 5 km in the horizontal and 100 m in the vertical. This resulted in approximately 23 000, 32 000 and 36 000 receptor points in 2012–2014, respectively. At each receptor point, 500 particles were released and traced backwards in time for five days. At each hour backwards in time, a surface sensitivity map is generated that represents the surface that influenced the measurement from that hour. These 120 surface sensitivity maps (24 h × 5 d) are added together to determine the cumulative 5-day footprint for a given receptor point. As described on P 8, lines 4–9 of the original submitted manuscript, the total footprint sensitivity associated with a vertical profile was determined by further binning these receptor points every 250 m and calculating the mean footprint at each height. The resulting profile of footprint sensitivities was then summed to determine the total surface influence on a given profile. The text in Sect. 3.4 has been improved to make this more clear.

5. P.6. Section 3.3 - Mixed layers: This relates very closely to the comment above. If mixed layer height (i.e. instantaneous local PBL height plus residual layer height) is used to derive the fluxes later on, how do you differentiate between the true PBL (which is in contact with the local surface) and the residual layer (which may represent the previous day's local PBL ventilation, or advection from a more distant regional source)? This is very confusing. The method states (page 6, line 26) that the authors use the integrated mixed-layer partial column concentration (not the vertically-resolved measurements) to calculate flux in Section 3.4. This must then surely convolve any local emissions (in the true local PBL) and non-local emission (in any residual layer). The authors later go

**to great lengths to show that any residual layer is not influence by long-range (non-regional) transport but this does not solve the problem of varying airmass histories for the true PBL versus the residual layer when these get clumped into a partial column for the purposes of the inversion. When this singular column concentration is used in STILT and coupled to the footprint described in Section 3.1 (and the issues alluded to in the previous comment), it seems impossible to deconvolve spatially-resolved flux with any true or traceable footprint sensitivity as the column represents an unknown mix of local and non-local surface contact. Again, I have a lot of sympathy (more than it may sound like) with the approach and doing the best job possible with adjoint models. But there is currently no awareness or clarity of these issues in the text which would alert the reader to the challenges and limitations in the approach.**

**Page 7 goes on to demonstrate that 50 percent of profiles were discarded from flux analysis because mixed layers could not be reliably separated or that PBL was "over-estimated" when compared to trace gas profiles. What does over-estimated mean here, and why are the remaining profiles more trustworthy for analysis than those retained? Discarding half of the dataset to select only those data that agree with a contrived and questionable model is a little worrying. I would have liked to know more as to why >50 percent of the dataset is at odds with the model/assumptions and to know what the sensitivity to including the data might be. Can you be confident that the way the remaining 50 percent has been retained hasn't led to some systematic bias in the data and model treatment?**

We agree that the partial column is influenced by both local sources in the boundary layer and regional sources in the mixed layer. The important point to note is that the spatial distribution of the footprint sensitivities of air sampled in the boundary and mixed layers will reflect local and regional sources, respectively, thus correctly attributing surface influences to the source regions.

The 273 profiles quoted in the main text represents every instance of the aircraft flying from the surface to >3 km or vice versa. The reviewer will appreciate that not every aircraft transit between these heights properly sampled the entire mixed layer and sufficient height into the free troposphere to properly determine the background CH4 mixing ratio. In addition, there are times when the atmospheric structure does not follow the structure of a classic textbook profile, with mixed layer heights from observations of water vapour, potential temperature and other trace gasses differing, either with each other or with the refractivity method. The cause of these inconsistencies would warrant a study of its own in boundary layer meteorology and is beyond the scope of our study. We therefore limit our calculations to profiles that are well-defined and where we believe we understand the dynamics.

As requested by the reviewer, we now include calculated net fluxes for all identified profiles in the Supplement. However, we do not believe that all of these calculated fluxes are representative of actual net surface emissions and therefore do not include flagged profiles in any of our calculations. The effect of excluding these profiles from the monthly mean cannot be determined. It should be noted that for all 273 profiles, the mean 95% confidence interval (C.I.) for an individual CH4 flux estimate was 8.1 mg m$^{-2}$ d$^{-1}$, ranging from 0.54–34 mg m$^{-2}$ d$^{-1}$ (2.5%–97.5% percentile). However, for the subset of 146 profiles used in our analysis, the average 95% C.I. was 5.2 mg m$^{-2}$ d$^{-1}$ and ranged from 0.6–15 mg m$^{-2}$ d$^{-1}$. As stated at the end of Sect. 3.4, we can say that based on the model footprint sensitivities, our sampling region is not skewed by excluding these profiles.

**6. P.8. Using CO as a tracer for combustion CH4 sources: What about pure fugitive emissions of thermogenic CH4 (where there is no combustion)? This could conceivably lead to an over-estimate of biogenic flux if the remaining profiles contain any significant non-biogenic CH4 from sources not co-emitted**

**with (potentially large fluxes of) CO. I see that only 9 of the profiles were discarded by this definition and the analysis of sensitivity by including them to derive a different flux is useful. This style of analysis starts to give the reader what they need to assess things.**

As the reviewer pointed out in the first specific comment, our calculations are only representative of net methane emissions in our study region, to which we attribute to biogenic emissions. These assumptions and other possible sources of methane are now discussed in Sects. 3.4 and 3.5. It should be noted that preliminary results by Floerchinger et al. (2016) on the North Slope of Alaska suggests that fugitive emissions from thermogenic sources are small compared to the net emissions and mostly confined to the region near Prudhoe Bay. Although these results are still preliminary, they indicate that our attribution of emissions to biogenic sources is reasonable.

**With this in mind, I have a few suggestions below to consider:**

**Suggested principal corrections:**

**7. The monthly-averages given may be hiding a wealth of useful data. A time series of biogenic flux (as an area-normalised quanitity – i.e. as biogenic flux per unit time per unit area) for each independent flux retrieval (I believe there are 68 of these?) would be useful. The posterior flux uncertainty (that STILT should yield as output) could be plotted as an error bar on each data point on such a plot.**

As discussed at the end of Sect. 3.4, 146 profiles are used to calculate the monthly means. As requested by the reviewer, we now include a time series of the flux calculated from each individual profile (Fig. S1) as well as the residuals from the

monthly mean (Fig. S2) in the Supplement. Our inversion method is based on simple linear regression and not a Bayesian analysis, therefore there is no posterior flux uncertainty from the model. Instead, we derive our uncertainties in the estimated flux by bootstrapping each element that goes into the calculation (observed mixing ratio, pressure, temperature and footprint sensitivity at each height bin, as well as background methane mixing ratio and mixed layer height). Over 500 iterations, we arrived at 95% confidence intervals which are shown as the uncertainties in Fig. S1. A detailed description of this analysis is now included in Sect. 3.7 in the revised manuscript.

**8. Error/uncertainty analysis: As discussed above, the current tolerance placed on the derived fluxes is meaningless and does not represent either systematic error (flux inversion uncertainty) or natural variability. The seasonal trend plotted in Figures 4 and 5 do not convince me that natural regional variability dominates the mean as this could simply be a manifestation of the changing northern hemispheric seasonal background and priors used. I would suggest that the posterior flux uncertainty of independent retrievals/footprints is used instead as this captures the uncertainty on each retrieval. And then, rather than a standard error on the mean flux (taken from the spread of the averaged inverted fluxes), which would clearly be an incorrect (and much reduced) error, I would recommend quoting the posterior flux uncertainty (calculated as an average of the posterior uncertainties across all inversion that contribute to the final monthly mean). It would be important to give the average of the posterior uncertainties (not their standard deviation or standard error), to yield a meaningful uncertainty on the monthly mean flux. Such an error will still convolve natural flux variability but at least it would be a more accurate measure of the systematic uncertainty in the method used. This should replace the shading (error bars) used in Figure 4.**

We thank the reviewer for this useful suggestion. We averaged the 95% confidence intervals as the uncertainty in an individual flux estimate to determine the uncertainty in the monthly mean as well as calculating a footprint weighted average. These results can now be seen for each month and on the growing season budget in Tables S5–S7. We chose instead to represent uncertainty in the monthly mean with the standard deviation weighted by the inverse of the 95% confidence intervals because they tended to result in larger uncertainties in the budget and we felt represented the variability associated with each month's measurements as well as the uncertainty associated with individual flux estimates. We now discuss these findings in the newly added Sect. 3.7 and have replaced the shading on Fig. 4 to reflect these new calculations.

**9 A table of the derived mean fluxes (and their corrected uncertainties) could be presented, which displays flux with and without the sensitivities/assumptions that have been used (i.e. masking seas and mountains, removing elevated CO profiles). These fluxes are currently in the body of the text, making it hard to compare them. And perhaps a new flux could be calculated where all 248 profiles are used in the inversion? A table would add (at a glance) the comparison between these sensitivities. This latter sensitivity test would then give the reader all the information they need to compare the information and make their own judgment about what they trust and the implications of the assumptions used.**

Tables S5–S7 in the Supplement now show the monthly mean net fluxes. As we discuss in the response to point 5 above, we do not believe that the excluded profiles provide meaningful estimates of net fluxes, and therefore do not include them in any monthly mean calculations. However, at the reviewer's suggestion, we now include flux estimates for all the individual profiles so that their quality can be judged by the reader.

**Technical corrections:**

**10. 1/ Title – hyphenate "regional-scale"**

**11. 2/ Abstract line 1: change to ". . .gas but its emissions. . .." Not "their".**

**12. 3/ P.1. Line 10: change to ". . .CH4 flux was. . ." or ". . .CH4 fluxes were. . ." - There seems to be some confusion between the use of singular and plural references throughout when referring to flux and fluxes, respectively. Please check as I won't list any further instances.**

**13. 4/ P.2 line 5: change to "40°N" and check throughout. "40N" is not acceptable.**

**14. P.2. Line 9: century should always be capitalized when referring to a specific century.**

**15. P.4. Line 6: add space to "195 K".**

**16. P. 8, lime 11: typo - change to "weighting".**

These have been corrected in the revised manuscript.

**References**

Floerchinger, C. R., McKain, K., Newberger, T., Handley, P., Wofsy, S. C., Sweeney, C., and Benmergui, J. S.: A methane budget from the North Slope of the Alaskan Brooks Range

including arctic tundra and proximate oil and gas operations, American Geophysical Union, San Francisco, CA, 12-16 December 2016, 2016.

Gerbig, C., Lin, J. C., Wofsy, S. C., Daube, B. C., Andrews, A. E., Stephens, B. B., Bakwin, P. S., and Grainger, C. A.: Toward constraining regional-scale fluxes of CO2 with atmospheric observations over a continent: 2. Analysis of COBRA data using a receptor-oriented framework, J. Geophys. Res., 108, n/a–n/a, doi:10.1029/2003JD003770, http://dx.doi.org/10.1029/2003JD003770, 4757, 2003.

---

## Author Comment (AC4) · 1 Jul 2017

The comment was uploaded in the form of a supplement:
https://www.atmos-chem-phys-discuss.net/acp-2017-72/acp-2017-72-AC4-supplement.zip